# Analysis of Total Column $CO_2$ and $CH_4$ Measurements in Berlin with WRF-GHG

Xinxu Zhao[1], Julia Marshall[2], Stephan Hachinger[3], Christoph Gerbig[2], Matthias Frey[4], Frank Hase[4], and Jia Chen[1]

[1]Electrical and Computer Engineering, Technische Unversität München, 80333 Munich, Germany
[2]Max Plank Institute of Biogeochemistry, 07745 Jena, Germany
[3]Leibniz Supercomputing Center (Leibniz-Rechenzenturm, LRZ) of Bavarian Academy of Sciences and Humanities, Bolzmannstr. 1, 85748 Garching, Germany
[4]Karlsruhe Institute of Technology (KIT), Institute of Meteorology and Climate Research (IMK-ASF), Karlsruhe, Germany

**Correspondence:** Jia Chen (jia.chen@tum.de)

**Abstract.** Though they cover less than 3 % of the global land area, urban areas are responsible for over 70 % of the global greenhouse gas (GHG) emissions and contain 55 % of the global population. A quantitative tracking of GHG emissions in urban areas is therefore of great importance, with the aim of accurately assessing the amount of emissions and identifying the emission sources. The Weather Research and Forecasting model (WRF) coupled with GHG modules (WRF-GHG) developed

for mesoscale atmospheric GHG transport, can predict column-averaged abundances of $CO_2$ and $CH_4$ ($XCO_2$ and $XCH_4$). In this study, we use WRF-GHG to model the Berlin area at a high spatial resolution of 1 km. The simulated wind and concentration fields were compared with the measurements from a campaign performed around Berlin in 2014 (Hase et al., 2015). The measured and simulated wind fields mostly demonstrate good agreement. The simulated $XCO_2$ shows quite similar trends with the measurement but with approx. 1 ppm bias, while a bias in the simulated $XCH_4$ of around 2.7 % is found. The

bias could be potentially the results of relatively high background concentrations, the errors at the tropopause height etc.. We find that an analysis using differential column methodology (DCM) works well for the $XCH_4$ comparison, as corresponding background biases then cancel out. From the tracer analysis, we find that the enhancement of $XCH_4$ is highly dependent on human activities. The $XCO_2$ signal in the vicinity of Berlin is dominated by anthropogenic behavior rather than biogenic activities. We conclude that DCM is an effective method for comparing models to observations independently of biases caused,

e.g., by initial conditions. It allows us to use our high resolution WRF-GHG model to detect and understand major sources of GHG emissions in urban areas.

# 1 Introduction

The share of greenhouse gas (GHG) emissions released from urban areas has continued to increase as a result of urbanization
(IEA, 2008; Kennedy et al., 2009; Parshall et al., 2010; IPCC, 2014). At present 55 % of the global population resides in urban
areas (UNDESA, 2014), a number that is projected to rise to 68 % by 2050 (UNDESA, 2018). Meanwhile urban areas cover
less than 3 % of the land surface worldwide (Wu et al., 2016), but consume over 66 % of the world's energy (Fragkias et al.,
2013), and generate more than 70 % of anthropogenic GHG emissions (Hopkins et al., 2016). Carbon dioxide ($CO_2$) emissions
from energy use in cities are estimated to comprise more than 75 % of the global energy-related $CO_2$, with a rise of 1.8 % per
year projected under business-as-usual scenarios between 2006 and 2030 (IEA, 2009). Methane ($CH_4$) emissions from energy,
waste, agriculture, and transportation in urban areas make up approximately 21 % of the global $CH_4$ emissions (Marcotullio
et al., 2013; Hopkins et al., 2016). As emission hotspots, urban areas therefore play a vital role in GHG mitigation. It is crucial
to find appropriate methods for understanding and projecting the effects of GHG emissions on urban areas, and for formulating
mitigation strategies.

There are two methods for the quantitative analysis of GHG emissions: the 'bottom-up' approach and the 'top-down' approach (Pillai et al., 2011; Caulton et al., 2014; Newman et al., 2016). The 'bottom-up' approach calculates emissions based
on activity data (i.e., a quantitative measure of the activity that can emit GHGs) and emission factors (Wang et al., 2009). This
approach has some uncertainty, e.g., on the national fossil-fuel $CO_2$ emission estimates, ranging from a few percent (e.g., 3 %-5
% for the US) to a maximum of over 50 % for countries with less resources for data collection and poor statistical framework
(Andres et al., 2012). The considerable uncertainties are caused by the large variability of source-specific and country-specific
emission factors and the incomplete understanding of emission processes (Montzka et al., 2011; Bergamaschi et al., 2015).
These uncertainties grow larger at sub-national scales, when estimating the disaggregation of the national annual totals in
space and time. The 'top-down' approach can not only provide estimated global fluxes, but also verify the consistency and assess the uncertainties of bottom-up emission inventories (Wunch et al., 2009; Montzka et al., 2011; Bergamaschi et al., 2018).
However, it is hard to quantify the statistical errors attached to both atmospheric observations and prior knowledge about the
distribution of emissions and sinks (Cressot et al., 2014).

McKain et al. (2012) suggested that column measurements can provide a promising route to improving the detection of $CO_2$
emitted from major source regions, possibly avoiding extensive surface measurements near such regions. Such measurements,
i.e. measurements of concentration averaged over a column of air, are performed to help to disentangle the effects of atmospheric mixing from the surface exchange (Wunch et al., 2011) and decrease the biases associated with estimates of carbon
sources and sinks in atmospheric inversions (Olsen and Randerson, 2004). Compared to surface values, urban enhancements
in columns are less sensitive to boundary-layer heights (Wunch et al., 2011; McKain et al., 2012; Kivi and Heikkinen, 2016)
and column observations have the potential to mitigate mixing height errors in an atmospheric inversion system (Gerbig et al.,
2008). Atmospheric GHG column measurements combined with inverse models are thus a promising method for analyzing
GHG emissions, and can be used to analyze their spatial and temporal variability (Ohyama et al., 2009; Pillai et al., 2011;
Ostler et al., 2016; Kivi and Heikkinen, 2016).

In order to focus the 'top-down' approach on concentration differences caused by local and regional emission sources, and in particular to quantify urban emissions, the differential column methodology (DCM) was proposed. It evaluates differences between column measurements at different sites. Chen et al. (2016) applied the DCM using compact Fourier Transform Spectrometers (FTS) EM27/SUN (Bruker Optik, Germany) and demonstrated the capability of differential column measurements for determining urban and local emissions in combination with column models. Citywide GHG column measurement campaigns have been carried out, e.g., in Boston (Chen et al., 2013), Indianapolis (Franklin et al., 2017), San Francisco, Berlin (Hase et al., 2015), and Munich (Chen et al., 2018). However, only a few studies have combined differential column measurements with high-resolution models. Toja-Silva et al. (2017) simulated the column data at upwind and downwind sites of a gas-fired power plant in Munich using the Computational Fluid Dynamic model (CFD) and compared them with the column measurements. Viatte et al. (2017) quantified $CH_4$ emissions from the largest dairies in the southern California region, using four EM27/SUNs in combination with the Weather Research and Forecasting model (WRF) in large-eddy simulation mode. Vogel et al. (2019) deployed five EM27/SUN in the Paris metropolitan area and analyzed the data with the atmospheric transport model framework CHIMERE-CAMS.

This paper carries out a quantitative analysis of GHG for the Berlin area in combination with DCM. We utilize the mesoscale WRF model (Skamarock et al., 2008) coupled with GHG modules (WRF-GHG) (Beck et al., 2011) at a high resolution of 1 km. The aim is to assess the precision of WRF-GHG and to provide insights on how to detect and understand sources of GHGs ($CO_2$ and $CH_4$) within urban areas. WRF is a numerical weather prediction system and can be used for both atmospheric research and operational forecasting, on a mesoscale range from tens of meters to thousands of kilometers, cf. e.g. (Chen et al., 2011). To produce high-resolution regional simulations of atmospheric $CH_4$ passive tracer transports, WRF was coupled with the Vegetation Photosynthesis and Respiration module (WRF-VPRM) (Ahmadov et al., 2007). WRF-VPRM has been widely employed in several studies, in which both the generally good agreement of the simulations with measurements and model biases have been assessed in detail (Ahmadov et al., 2009; Pillai et al., 2011, 2012; Kretschmer et al., 2012). Biogenic carbon fluxes given by the VPRM model tend to underestimate urban ecosystem carbon exchange, owing to the incomplete understanding of urban vegetation, and to conditions related to urban heat islands and altered urban phenology (Hardiman et al., 2017). WRF-VPRM was later extended to WRF-GHG (Beck et al., 2011), which can simulate the regional passive tracer transport for GHGs ($CH_4$, $CO_2$ and carbon monoxide (CO)). Relatively few studies using WRF-GHG have been published as yet. Pillai et al. (2016) utilized a Bayesian inversion approach based on WRF-GHG at a high spatial resolution of 10 km for Berlin to obtain anthropogenic $CO_2$ emissions, and to quantify the uncertainties in retrieved anthropogenic emissions related to instruments (e.g. CarbonSat) and modelling errors. Pillai et al. (2016) was an observation system simulation experiment, based on synthetic data rather than real observations as in our case. In the present paper, our focus is on a high-resolution (1 km) study of both $CO_2$ and $CH_4$ in Berlin, and assessing the performance of WRF-GHG through comparing the simulated wind and concentration fields to observations from wind stations and ground-based solar-viewing spectrometers. Then DCM is tested as a proper approach for model analysis, which can cancel out the bias from initialization conditions and highlight regional emission tracers. The simulation workflow is also adapted to this purpose where needed. This study is the fundamental study of the WRF-GHG mesoscale modeling framework in urban areas.

The total annual $CO_2$ emissions of Berlin (21.3 million tonnes in 2010) approximately correspond to those of Croatia, Jordan or the Dominican Republic (Reusswig and Lass, 2014). With its strong regulatory influence as a 'state' within Germany, and a strongly supportive policy, Berlin has already transformed itself into a climate-friendly city in which $CO_2$ emissions have

90 been reduced by a third compared with 1990 levels, aiming for carbon neutrality by 2050 (Homann, 2018). Berlin therefore needs to assess and identify the emission sources accurately at the current stage, to provide solid scientific support for the selection of mitigation options. Additionally, Berlin is an ideal pilot case for developing and testing simulations because the city is relatively isolated from other large cities with high emissions, such that anthropogenic GHG anomalies around Berlin can confidently be attributed to the city itself.

The major goals of our work in this context are: (1) to simulate high-resolution (1 km) $CO_2$ and $CH_4$ concentrations for Berlin using WRF-GHG, attributing the changes in concentrations to different emission processes; (2) to compare the simulation outputs with the observations from a column measurement network in Berlin (Hase et al., 2015), assessing the precision of WRF-GHG; (3) to use DCM in the simulation analysis, testing the feasibly of this approach. The structure of this paper is as follows: The model with its domain and external data sources are described in Sect.2. A comparison analysis for wind

fields and concentration fields is presented in Sect.3, and $CO_2$ and $CH_4$ concentrations related to different processes (e.g., the anthropogenic component) are discussed. DCM for the comparison of concentration fields and the tracer analysis is presented and discussed in Sect.4. Section 5 provides the discussion and summary of this study.

## 2   WRF-GHG Modeling System

As mentioned in Sect.1, we use the WRF model Version 3.2 coupled with GHG modules to quantify the uptake and emission

of atmospheric GHGs around Berlin at a high resolution of 1 km. WRF follows the fully compressible nonhydrostatic Euler equations (Skamarock et al., 2005, 2008) and is based on the actual meteorological data in this case study. The meteorological initial conditions and lateral boundary conditions were taken from the Global Forecast System (GFS) model reanalysis in which in-situ measurements and satellite observations have been assimilated. Tracers in WRF-GHG are transported online in a passive way, i.e. without any chemical loss or production, when the tracer transport option is used (Ahmadov et al., 2007;

Beck et al., 2011). As shown in Fig.1, three domains are set up here, whose dimensions are $70 \times 50$ horizontal grid points with a spacing of 9 km for the coarsest domain (d01), 3 km for the middle domain (d02) and 1 km for the innermost domain (d03). WRF uses a terrain-following hydro-static pressure vertical coordinate (Skamarock et al., 2008). In our case, 26 vertical levels are defined from the surface up to 50 hPa, 14 of which are in the lowest 2 km of the atmosphere. The innermost domain, d03, envelops all five measurement sites (see Sect.3.1) to assess the simulation by comparing with the measured data. Berlin lies

in the North European Plain on flat land (crossed by northward-flowing watercourses), which avoids the vertical interpolation problems caused by topography differences (Fig.1). The Lambert conformal conic projection is selected as map projection. The simulated time span is from 18 UTC on 30[th] June to 00 UTC on 11[th] July in 2014. The description of the workflow for running WRF-GHG can be found in Appendix A.

The meteorological fields are obtained from the Global Forecast System model (GFS) at a horizontal resolution of 0.5°
with 64 vertical layers and a temporal resolution of 3 hours (as available via the NOAA-NCDC/NCEI, www.ncdc.noaa.gov).
The GFS uses hydrostatic equations for the prediction of atmospheric conditions, and its output includes large amounts of
atmospheric and land-soil variables, wind fields, temperature, precipitation and soil moisture etc. The initial and lateral bound-
ary conditions for our WRF-GHG concentration fields are implemented using Copernicus Atmosphere Monitoring Service
(CAMS) data (Agusti-Panareda et al., 2017). CAMS provides the estimated mixing ratios of $CO_2$ and $CH_4$ with a spatial reso-
lution of 0.8° on 137 vertical levels, with a temporal resolution of 6 hours (as available via https://atmosphere.copernicus.eu).

The simulation of $CO_2$ and $CH_4$ fluxes with different emission tracers in WRF-GHG is based on flux models and emission
inventories which are either already implemented inside the model modules ('online' calculation) or constitute external datasets
('offline' calculation). The flux values from external emission inventories are converted into atmospheric concentrations and
added to the corresponding tracer variables. In combination with the background concentration fields for $CO_2$ and $CH_4$ that
refer to the $CO_2$ and $CH_4$ values without any sources and sinks in the targeted domain, the tracer contributions are summed up
to obtain the total concentrations, as

$$CO_{2,\text{total}} = CO_{2,\text{bgd}} + CO_{2,\text{VPRM}} + CO_{2,\text{anthro}} + \Delta CO_2$$
$$CH_{4,\text{total}} = CH_{4,\text{bgd}} + CH_{4,\text{anthro}} + CH_{4,\text{soil}} + \Delta CH_4$$

(1)

where $CO_{2,\text{total}}$ and $CH_{4,\text{total}}$ represent the total $CO_2$ and $CH_4$, $CO_{2,\text{bgd}}$ and $CH_{4,\text{bgd}}$ are the background $CO_2$ and $CH_4$, $CO_{2,\text{anthro}}$
and $CH_{4,\text{anthro}}$ stand for the changes in $CO_2$ from the anthropogenic emissions, $CO_{2,\text{VPRM}}$ is the change in $CO_2$ from the biogenic
activities and $CH_{4,\text{soil}}$ is the change in $CH_4$ from soil uptake, $\Delta CO_2$ and $\Delta CH_4$ are the tiny computational errors for $CO_2$ and
$CH_4$ described in detail in Appendix B. In the transport process, the relationship shown in Eq.1 holds for each vertical level.

The biogenic $CO_2$ emission is calculated online using VPRM (Mahadevan et al., 2008), in which the hourly Net Ecosystem
Exchange (NEE) of $CO_2$ reflects the biospheric fluxes between the terrestrial biosphere and the atmosphere, estimated by
the sum of Gross Ecosystem Exchange (GEE) and Respiration. VPRM in WRF-GHG calculates biogenic fluxes initialized
by vegetation indices (land surface water index (LSWI), enhanced vegetation index (EVI), etc..) from the MODIS satellite (as
available via https://modis.gsfc.nasa.gov/). The SYNMAP vegetation classification at a resolution of 1 km and 8-day reflectance
data from the MODIS satellite at 500- 1000 m spatial resolution (depending on the wavelength band) are aggregated onto the
Lambert Conformal Conic (LCC) projection within the VPRM preprocessor. Then, the data including these high-solution
vegetation indexes at a resolution of 1 km are available on the model domains.

We use the external dataset Emission Database for Global Atmospheric Research Version 4.1 (EDGAR V.4.1) for the anthro-
pogenic fluxes in our study. EDGAR V.4.1 provides annually varying global anthropogenic GHG emissions and air pollutants
at a spatial resolution of 0.1° (Muntean et al., 2014; Janssens-Maenhout et al., 2015), whose source sectors include industrial
processes, on-road and off-road sources in transport, large-scale biomass burning and other anthropogenic sources (Saikawa
et al., 2017). Here we apply time factors for seasonal, weekly, daily and diurnal variations defined by the time profiles published
on the EDGAR website (http://themasites.pbl.nl/tridion/en/themasites/edgar/documentation/content/Temporal-variation.html);
however, considerable uncertainties are to be expected in applying these time factors. This temporal variation set is derived

based on western European data such that the representativity for other European countries and even other world regions may be quite poor. The coarse emission fluxes used for the initialization of the anthropogenic tracer in WRF-GHG can cause problems when locating emission points within the high-resolution model grid, and can weaken the impact from the real high emission hot spots in the fine domain of our study. The chemical sink for atmospheric $CH_4$ (e.g., photo-chemistry in the stratosphere) can be ignored in the model owing to its relatively long lifespan ($9.5 \pm 1.3$ year, Holmes (2018)), the small-scale domains, and the limited simulation period (10 days) in our case.

## 3 Model Analysis and Model-Measurement Comparison

### 3.1 Description of Measurement Sites

The measurement campaign used to compare with WRF-GHG in this paper was performed from 23[rd] June to 11[th] July 2014 in Berlin using five spectrometers (Hase et al., 2015). It allows us to both test the precision of WRF-GHG (Sect.3) and verify differential column methodology (DCM) as our analytic methodology (Sect.4). In their measurement campaign, Hase et al. (2015) used five portable Bruker EM27/SUN Fourier Transform Spectrometers (FTS) for atmospheric measurements based on solar absorption spectroscopy. Five sampling stations around Berlin were set up, four of which (Mahlsdorf, Heiligensee, Lindenberg and Lichtenrade) were roughly situated along a circle with a radius of 12 km around the center of Berlin. Another sampling site was closer to the city center and located inside the Berlin motorway ring at Charlottenburg (Fig.6). Detailed information on this measurement campaign is given in Hase et al. (2015) and Frey et al. (2015) provides additional details on the calibration of the spectrometers, precision and instrument-to-instrument biases.

### 3.2 Comparison of Wind Fields at 10 meters

Winds have a strong impact on the vertical mixing of GHGs and a direct influence on their atmospheric transport patterns. Hence, we firstly compare the wind speeds and wind directions obtained from WRF-GHG to the measurements, such that deviations between the simulated and measured wind fields are assessed. The wind measurements are not exactly co-located with the spectrometers mentioned in Sect.3.1, but rather are located at three sampling sites (Tegel, Schönefeld and Tempelhof, respectively) and measure at a height of 10 meters above the ground. The simulated wind speed at 10 meters ($ws_{10m}$) and wind direction at 10 meters ($wd_{10m}$) are calculated following the equations,

$$
\begin{aligned}
ws_{10\text{m}} &= \sqrt{u_{10\text{m}}^2 + v_{10\text{m}}^2} \\
wd_{10\text{m}} &= \arctan \frac{v_{10\text{m}}}{u_{10\text{m}}}
\end{aligned}
\tag{2}
$$

where $u_{10\text{m}}$ and $v_{10\text{m}}$ are the components of the horizontal wind, towards the east and north respectively, which can be obtained from WRF-GHG output files.

Figure.2 shows the comparisons of wind speeds (Fig.2(a)) and wind directions (Fig.2(b)) between simulations and observations at 10 meters from 1[st] July to 10[th] July and the model-measurement differences. EM27/SUN only operates in the daytime

when there is sufficient sunlight (the detailed description of the instrument can be found in Gisi et al. (2012), Frey et al. (2015) and Vogel et al. (2019)). The instrumental working periods are marked by gray shaded boxes in Fig.2. The measured (dashed lines) and simulated (solid) wind speeds (Fig.2(a)) at 10 meters show similar trends and demonstrate relatively good agreement over the 10-day time series with a root mean square error (RMSE) of $0.9247\,\mathrm{m/s}$. Large uncertainties in wind speeds are found to appear always with the lower wind speeds, mostly at night. In terms of wind directions at 10 meters, we observe that the simulated wind directions show similar but slightly underestimated fluctuations (Fig.2(b)), which result in a RMSE of $60.8328°$. Larger uncertainties in wind directions always exist during the low wind speed periods (Fig.2(a)&(b)). During the instrumental working period (within the daytime), the simulations fit better with the measurements with relatively lower RMSEs $0.6928\,\mathrm{m/s}$ for wind speeds and $41.4793°$ for wind directions. We find that the measured wind fields (both wind speeds and wind directions) have more fluctuations, compared to the simulations. This could be caused by real fast wind changes which the model, simulating a somewhat idealized environment, is not able to capture. To be specific, local turbulence given by urban canopy, buildings etc. are not represented well in the model.

### 3.3 Comparison of pressure-weighted column-averaged concentrations

In the following, we use the measured concentration fields to compare with the simulated fields. An FTS EM27/SUN can measure the column-integrated amount of a tracer through the atmospheric column with excellent precision, yielding the column-averaged dry-air mole fractions (DMFs) of the target gases (Chen et al., 2016; Hedelius et al., 2016). The measured DMFs of $CO_2$ and $CH_4$ are denoted by $XCO_2$ and $XCH_4$. Hase et al. (2015) used constant a priori profile shapes in the retrievals of measurements.

When comparing remote sensing observations to model data (or also data sets from different remote sensing instruments to one another), limitations of the instruments in reconstructing the actual atmospheric state need to be taken into account. In general, this requires the a-priori profile which was used for the retrieval and the averaging kernel matrix, which specifies the loss of vertical resolution (fine vertical details of the actual trace gas profile cannot be resolved) and limited sensitivity (e.g. Rodgers and Connor (2003)). In the case of EM27/SUN, the spectrometers used in the network offer only a low spectral resolution of $0.5\,\mathrm{cm^{-1}}$. Therefore, performing a simple least squares fit by scaling retrieval of the a-priori profile is generally appropriate. In this case, there is no need to specify a full averaging kernel matrix, instead, the specification of a total column sensitivity is sufficient. The total column sensitivity is a vector (being a function of altitude), which specifies to which degree an excess partial column superimposed on the actual profile at a certain input altitude is reflected in the retrieved total column amount. This sensitivity vector is a function of solar zenith angle (and ground pressure), mainly due to the fact that the observed signal levels in different channels building the spectral scene used for the retrieval are shaped by a mixture of weaker and stronger absorptions. (If all spectral lines in the spectral scene would be optically thin and too narrow to be resolved by the spectral measurement, the sensitivity would approach unity throughout.

In order to ensure measurement qualities and enough sample points for further concentration comparisons, we select five measurement dates (1[st], 3[rd], 4[th], 6[th] and 10[th] July) with relatively good measurement qualities (from fair '++' to very good '++++') based on Hase et al. (2015). The pressure-dependent column sensitivities for $CO_2$ (Fig.3(b)) and $CH_4$ (Fig.3(c)) are

derived from measurements performed in Lindenberg on 4$^{th}$ July (the best-quality day in terms of measurements). Details about the measurements can be found in Hase et al. (2015) and Frey et al. (2015). The shape and values of the column sensitivities from Karlsruhe closely resemble the results of Hedelius et al. (2016) in Pasadena. As depicted in Fig.3(a), the solar zenith angles (SZAs) are almost identical for each day in our study (at each hour), rendering the shape of column sensitivities (at a specific hour of the day) practically independent of the measurement date. The column sensitivities for 4$^{th}$ July (Fig.3(b,c)),

are taken as a basis for our smoothing process below. The a-priori $CO_2$ and $CH_4$ profiles have been taken from the Whole Atmosphere Community Climate model (WACCM) Version 6. A smoothed profile for a target gas $G$ is then obtained as Eq.3 in (cf. Vogel et al., 2019),

$$G^s = K * G + (I - K) * G^p \qquad (3)$$

where $G$ is the modelled profile from WRF-GHG, $I$ is the identity matrix, $K$ is a diagonal matrix containing the averaging

kernel, and $G^p$ is the a-priori profile.

In order to compare the simulated smoothed concentration fields with the observations, the simulated smoothed pressure-weighted column-averaged concentration for a target gas $G$ ($XG$) is calculated as,

$$\Delta p(i) = \frac{P(i) - P(i+1)}{P_{\text{sf}} - P_{\text{top}}} \rightarrow XG = \sum_{i=1}^{n} \Delta p(i) \times G^s(i) \qquad (4)$$

Here, $\Delta p_i$ is the proportional to the differences of the pressure values $P(i)$ at the bottom and $P(i+1)$ at the top of the $i^{th}$

vertical grid cell; $P_{\text{top}}$ and $P_{\text{sf}}$ represent the hydrostatic pressures at the top and at the surface of the model domain, and $G^s(i)$ stands for the simulated concentration of the target gas $G$ at the $i^{th}$ vertical level.

In Figures.D1 and D2 of Appendix D, we compare the simulated $XCO_2$ and $XCH_4$ with and without smoothing. The simulated concentrations are only slightly enlarged after smoothing, approximately 1-2 ppm for $XCO_2$ and 2 ppb for $XCH_4$, while the variations are mostly not changed. Compare to the period with lower SZAs (at noon), the smoothed values in the

morning and afternoon with higher SZAs hold relatively larger enlargements.

Figure.4(a) shows the measured and modelled variations of $XCO_2$ and $XCH_4$ for these five days. Compared to the measurements, the smoothed simulated pressure-weighted column-averaged concentrations for $CO_2$ ($XCO_2$) show quite similar trends but with approx. 1-2 ppm bias, indicated by a RMSE of 1.2534 ppm. The simulated $XCO_2$ are overestimated for 1$^{st}$, 3$^{rd}$ and 4$^{th}$ July while on 6$^{th}$ and 10$^{th}$ July, the model is underestimated, which could be the result of uncertainties from the coarse

anthropogenic surface emission fluxes, background concentrations from CAMS (Sembhi et al., 2015), and the ignorance of the influence from the line of the sun sight.

Figure.4(b) shows the comparison of the pressure-weighted column-averaged concentrations for $CH_4$ ($XCH_4$) between observations and smoothed simulations on the five selected dates (1$^{st}$, 3$^{rd}$, 4$^{th}$, 6$^{th}$ and 10$^{th}$ July). We find that there is an approximate offset of 50-60 ppb between observations and models (RMSE = 58.1082 ppb). The simulated $XCH_4$ is around 1860

ppb while the measured value is around 1810 ppb which is comparable to the values (1790-1810 ppb) observed at two Total Carbon Column Observing Network (TCCON) measurement sites in June and July 2014, Bremen in Germany (Notholt et al., 2014) and Bialystok in Poland (Deutscher et al., 2014). This bias of the simulated $XCH_4$ seems to be constant (around 2.7 %)

each day. Thus, we introduce an offset applied to all sites for each simulation date to compare the model and the measured data, effectively removing the bias, which we attribute to a too high background $XCH_4$. The daily offset is assumed to be the

difference between the simulated and measured daily mean $XCH_4$. After applying the daily offset, the measured $XCH_4$ shows a somewhat better agreement and a similar trend but with larger variability, compared to the simulation (RMSE = 3.1690 ppb). The smaller variations from the simulation results can, e.g., be caused by the error from the spatial-temporal treatment of emission maps, underestimated emissions from anthropogenic activities, the coarse wind data and/or the smoothing of actual extreme values in the simulation.

A major offset in modelled $CH_4$ concentration fields could potentially be attributed to the errors in the troposphere height and a general offset from CAMS. In the $CH_4$ vertical concentrations profile, we find that the typical sharp decrease occurs at the tropopause height. Tukiainen et al. (2016) also find the similar sharp decrease when using the AirCore to retrieve atmospheric $CH_4$ profiles in Finland. During the simulation, the background concentration values of CAMS are directly fitted to the WRF pressure axis, without considering the actual tropopause height, thus this could cause some error. An illustration of the vertical

distribution for $CH_4$ is provided in Appendix C. In contrast, the $CO_2$ vertical distribution shows a quite flat decrease with the increase of pressure and there is no need to consider the tropopause height during the grid treatment in the vertical layer. In terms of CAMS, the reports from Monitoring Atmospheric Composition and Climate (MACC) described that CAMS has a bias and RMSE (approx. 50 ppb) in each part of the world, compared to the Integrated Carbon Observation System (ICOS) observations in 2017 (Basart et al., 2017). Galkowski et al. (2019) also mentioned one $CH_4$ offset (approx. 30 ppb within

troposphere), when initializing the concentration fields using CAMS. Apart from these two major potential reasons for the bias, the influence from the inaccurate simulated planetary boundary layers and the shape of the constant a priori profile used for the retrievals could both potentially contribute to the discrepancies for the concentration fields. Due to the lack of fine measured vertical concentration profiles, it is not easy to quantify these errors and attribute these potential reasons to this 2.7% error quantitatively. Thus, a DCM-based analysis is presented in Sect.4, aiming at eliminating the bias from these relatively

high initialization values for $CH_4$ and making it easier to assess WRF-GHG results with respect to the measurements.

## 3.4  Contributions of different sources and sinks to the total signal: Individual Emission Tracers

As described in Sect.2, the various flux models implemented in WRF-GHG are advected as separate tracers, making it possible to distinguish the signals in concentration space for different source and sink categories for $CO_2$ and $CH_4$ (Beck et al., 2011). Berlin is located in an area of low-lying, marshy woodlands with a mainly flat topography (Kindler et al., 2018). There is no

wetland in Berlin according to the MODIS Land Cover Map (Friedl et al., 2010). The land covered by forests, green and open spaces (e.g., farmlands, parks, allotment gardens) accounts for 35 % of the total area in Berlin (SenStadtH, 2016). Additionally, eleven power plants are currently being operated in Berlin, eight of which have a capacity over 100 MW (Fraunhofer-Gesellschaft, 2018). In accordance with the geographical characteristics of the district and potential emission sources in Berlin, we focus on understanding the major emissions caused by vegetation photosynthesis and respiration ($XCO_{2,VPRM}$) as well as

anthropogenic activities ($XCO_{2,anthro}$) for $CO_2$, and by soil uptake ($XCH_{4,soil}$) as well as human activities ($XCH_{4,anthro}$) for $CH_4$.

As an instructive example of an analysis involving these tracers, we look at the diurnal cycle of contributions from the different tracers mentioned above in Charlottenburg (Fig.5). The mean values, averaged over 9 days (from 2$^{nd}$ to 10$^{th}$ July) as well as a 95 % confidential interval calculated in the averaging process are shown in Fig.5. Figure.5(a) clearly shows a decline during the day and a rise at night in the $XCO_2$ enhancement over the background (blue: $XCO_{2,\text{total}}$ - $XCO_{2,\text{bgd}}$), with a maximum decrease over the course of the day of around 2 ppm. The $XCO_2$ enhancement over the background reaches its daily peak during morning rush hour (07 UTC). The morning peak corresponds to $XCO_2$ changes from human activities, depicted by the black line from 04 UTC to 07 UTC (marked by a red square in Fig.5(a)). Before the evening rush hour (16 UTC), $XCO_2$ over the background then decreases, owning to biogenic uptake. Beginning in the evening, values increase again. The fluctuation in the evening (17 UTC – 19 UTC) are dominated by $XCO_2$ enhancements from human activities while the substantial rise from 19 UTC onward is generated by the VPRM tracer, specifically the accumulation of the vegetation respiration in the evening.

XCO2 is weaker compared to the strong biogenic uptake. To further highlight the role of anthropogenic activities in $XCO_2$ changes within the urban area, DCM is applied in Sect.4. More specifically, we will use downwind-minus-upwind column differences of $CO_2$ ($\Delta XCO_2$) to describe the $XCO_2$ enhancement over an upwind site, as the difference between the downwind and upwind sites can be attributed to urban emissions.

Turning to $XCH_4$ in Fig.5(b), we plot the variations of the mean hourly contributions from the anthropogenic (black line: $XCH_{4,\text{anthro}}$) and soil uptake tracer (blue: $XCH_{4,\text{soil}}$) in Charlottenburg. The contributions by anthropogenic activities fluctuate slightly around 2 ppb in the morning and at noon; then a peak occurs at the start of the evening rush hour (16 UTC). After 18 UTC, values clearly decrease, reaching approximately 2 ppb. From 21 UTC $XCH_4$ stabilizes, exhibiting only moderate fluctuations. The $XCH_4$ enhancement above the background (green: $XCH_{4,\text{total}}$ - $XCH_{4,\text{bgd}}$) depends largely on the $XCH_4$ contributions by human activities. The changes in concentrations caused by the soil uptake tracer (blue), whose values fluctuate between 0.001 ppb and 0.01 ppb, have almost no influence on the variation of the $XCH_4$ enhancement over background in the urban area.

## 4 Model Analysis using Differential Column Methodology

### 4.1 Comparison of differential column concentrations

The differential column methodology (DCM) can be employed to detect and estimate local emission sources within an area, based on calculated concentration differences between downwind and upwind sites (Chen et al., 2016). The difference ($\Delta XG$) of a specific gas $G$ in column-averaged DMFs across the downwind and upwind sites is defined as,

$$\Delta XG = XG_{\text{downwind}} - XG_{\text{upwind}} \tag{5}$$

where $XG_{\text{downwind}}$ and $XG_{\text{upwind}}$ represent the column-average DMFs at the downwind and upwind sites.

In this study, DCM is applied to measurements and models in the spirit of a post-processing analysis. This approach is not only useful to cancel out the bias of the simulated $XCH_4$ (see Sect.3.3), but also to assess the role of anthropogenic activities in $XCO_2$ changes more appropriately.

A necessary prerequisite for DCM is distinguishing the upwind and downwind sites among all five sampling sites. Wind direction thus plays a pivotal role in the calculation of the downwind-minus-upwind column differences. In this study, the hourly simulated vertical-averaged wind directions are assumed as a standard to classify the sites into downwind and upwind sites. The tracer transport calculations in the first few hours are not stable in WRF-GHG. Thus, we select 3rd, 4th, 6th and 10th as our targeted dates.

Table.1 shows the daily average wind directions with standard derivations and the details on the downwind and upwind sites for these four target dates. West wind is the prevailing wind direction on 3rd. That is to say, Mahlsdorf and Lindenberg are downwind sites, and the upwind sites corresponding to these are Charlottenburg and Heiligensee. While the wind on 10th July is northeasterly and the combination of downwind and upwind sites are selected to be opposite of the ones on 3rd July. The prevailing wind on 4th and 6th are easterly. The upwind site is Lichtenrade, and the corresponding downwind sites are 
Heiligensee and Lindenberg. Based on the selection of downwind and upwind sites shown in Table.1 and Eq.5, differential column concentrations ($\Delta XCH_4$) are, therefore, respectively, calculated as:

West Wind (3rd July) : $\Delta XCH_4 = (XCH_4^{\text{Mahlsdorf}} + XCH_4^{\text{Lindenberg}})/2 - (XCH_4^{\text{Charlottenburg}} + XCH_4^{\text{Lichtenrade}})/2$ (6)

North Wind (4th and 6th July) : $\Delta XCH_4 = (XCH_4^{\text{Heiligensee}} + XCH_4^{\text{Lindenberg}})/2 - XCH_4^{\text{Lichtenrade}}$ (7)

Northeast Wind (10th July) : $\Delta XCH_4 = (XCH_4^{\text{Charlottenburg}} + XCH_4^{\text{Lichtenrade}})/2 - (XCH_4^{\text{Mahlsdorf}} + XCH_4^{\text{Lindenberg}})/2$ (8)

Figure.7 depicts the variations of the wind fields (wind speeds and wind directions) and $\Delta XCH_4$ (corresponding to Eq.6, 7 and 8) on 3rd, 4th, 6th and 10th July. As depicted in the left column of Fig.7, the hourly vertical-mean simulated wind speeds and directions at downwind and upwind sites are homogeneous. Thus, it is reasonable to use the daily mean wind directions as the standard for the selection of downwind and upwind sites. The general trends in the simulated $\Delta XCH_4$ values, shown in the right column of Fig.7, seem to be roughly reproduced by the observations but slightly overestimated, with a RMSE of 
1.3895 ppb.

Yet, DCM as presented here has potential to highlight the role of anthropogenic activities, which we demonstrate applying it to $CO_2$ tracers in the simulation. Thus, the analysis on anthropogenic and biogenic tracers for $CO_2$ will be specially prominent here. As described above, we continue to take 3rd, 4th, 6th and 10th July as examples (see the left column of Fig. 8).

With the left column of Fig.8, the variations of $\Delta XCO_2$ (corresponding to Eq.6, 7 and 8) on 3rd, 4th, 6th and 10th are shown. 
In contrast to $XCO_2$ values (Sect.3.4, Fig.5(a)), the simulated $\Delta XCO_2$ (Fig.8, left column, blue lines) is not so much influenced by the $XCO_2$ changes from the VPRM tracer (Fig.8, left column, green), but more closely follows the $XCO_2$ changes from anthropogenic activities (Fig.8, left column, red). With DCM, the role of human activities in $XCO_2$ changes is highlighted and the strong effect from the biogenic component is canceled out. The $\Delta XCO_2$ measurements (Fig.8, left column, black) show similar trends as the simulation with a RMSE of 0.2973 ppm.

To further understand the differences of $\Delta XCO_2$ and $\Delta XCH_4$ between measurements and simulations (see Fig.7, right column and Fig.8, left column), the comparison of hourly mean $\Delta XCO_2$ and $\Delta XCH_4$ values for these four targeted dates is illustrated in the right column of Fig.8. Due to the restriction of measured wind information, we illustrate the differences of simulated and measured wind directions at 10 meters (i.e. Fig.2(b)) with respect to the hourly mean $\Delta XCO_2$ and $\Delta XCH_4$. We find that the real hourly mean $\Delta XCO_2$ and $\Delta XCH_4$ values are generally higher than the simulated values. Extreme points

are colored by red and blue in the right column of Fig.8, standing for large differences between measured and simulated wind directions at 10 meters. We see that a large difference of wind directions is a necessary but insufficient condition for the bias of $\Delta XCO_2$ and $\Delta XCH_4$ between measurements and simulations. In future studies, this is suggested to be verified further.

    We conclude that DCM, as applied in this plot, reduces the model bias caused by the simulation initialization, but introduces unpleasant effects which may be attributed to errors in the assumed or simulated wind directions.

## 4.2   Comparison between differential column concentrations and modeling results after the elimination of wind influence

As described in Sect.4.1, the wind direction impacts the distinction between downwind and upwind sites for DCM. Devising meaningful and accurate recipes for determining the wind directions is not easy, sometimes resulting in mixed-quality results (of Sect.4.1). Our simulated output provides the hourly wind and concentration fields. The instruments measure the concentra-

tion value every minute (Hase et al., 2015) We simply assume the wind direction to be a constant value within one hour (the hourly vertical-averaged values) in our calculation, also when it comes to selecting up- and downwind sites. This may create inaccuracies in the calculation of the measured $\Delta XCH_4$.

    In this section, we test replacing the upwind values in DCM by an all-site mean to provide a potential solution for the elimination of such problems while still applying the DCM. The mean of the column-averaged DMFs over all sampling sites

($\overline{XG}_{\text{specific site}}$) is assumed to be the background concentration within the entire urban region, replacing the $XCH_4$ at the upwind site. The differences between the specific site and the mean of all the sites for each gas $G$ ($\Delta\overline{XG}_{\text{specific site}}$) is then evaluated, i.e.

$$\Delta\overline{XG}_{\text{specific site}} = XG_{\text{specific site}} - \overline{XG}_{\text{all sites}} \tag{9}$$

where $XG_{\text{specific site}}$ is the column-averaged DMF at the respective sampling site.

We now test this form of DCM for the same four targeted dates (3rd, 4th, 6th and 10th July). The distance between any two sampling sites is around 25 km. The general trends of the simulated (Fig.9, blue lines) and measured (Fig.9, black) $\Delta\overline{XCH_4}$ appear to be more similar with a RMSE of $0.6698\,\text{ppb}$, compared to the comparison of $\Delta XCH_4$ in the right column of Fig.7 (RMSE=$1.3895\,\text{ppb}$). The measurement-model bias can be caused by underestimated emissions from anthropogenic activities, the smoothing of actual extreme values in the simulation and the ignorance of the line of the sun sight for the simulation. The

variations of the $XCH_4$ at the five different sampling sites on the same day are similar (Fig.9), but the measurements show more extreme values (e.g., 4th July), compared to the simulations. A further analysis in a future study is suggested to provide deeper insight on site-specific transport characteristics.

As a final point in our analysis, we focus on simulated $\Delta\overline{\mathrm{XCO_2}}$ values for these four target dates (Fig.10). The $\Delta\overline{\mathrm{XCO_2}}$ (blue line) on 3[rd], 4[th], 6[th] and 10[th] July in five sampling sites are mainly dominated by the $\mathrm{XCO_2}$ changes caused by the anthropogenic tracer (red), instead of the VPRM tracer (green). Compared to the left column of Figure.8, the red line and blue line in Fig.10 show a stronger similarity in their trends. With this form of DCM (compared to the original form Eq.5 in Sect.4.1), anthropogenic activities can be clearly shown to influence $\mathrm{XCO_2}$ within urban areas. Meanwhile, the $\Delta\mathrm{XCO_2}$ measurements (black) fit better with the simulation with a RMSE of $0.2333\,\mathrm{ppm}$, compared to the comparisons of $\Delta\mathrm{XCO_2}$ depicted in the left column of Fig.7 (RMSE=$0.2973\,\mathrm{ppm}$).

## 5 Discussion and conclusion

We used WRF-GHG to quantitatively simulate the uptake, emission and transport of $CO_2$ and $CH_4$ for Berlin with a high resolution of 1 km. The simulated wind and concentration fields were compared with observations from 2014. Then, differential column methodology (DCM) was utilized as a post-processing method for the $\mathrm{XCH_4}$ comparison and the $\mathrm{XCO_2}$ tracer analysis.

The measured and simulated wind fields at 10 meters mostly demonstrate good agreement but with slight errors in the wind directions. The simulated pressure vertical profile and the averaging kernel from the solar-viewing spectrometer (EM27/SUN) are used to obtain the smoothed pressure-weighted average concentration for further comparisons. The simulated $\mathrm{XCO_2}$ concentrations actually reproduce the observations well, but with approx. 1-2 ppm bias which can attribute to the coarse emission inventory, background concentrations from CAMS and the ignorance of the line of the sun sight for the simulation. Compared with the measured $\mathrm{XCH_4}$, some deviations can clearly be noted in the the simulated $\mathrm{XCH_4}$, mostly caused by the relatively high background concentration fields and the errors at the tropopause height. We discussed the diurnal variation of concentration components corresponding to the major emission tracers for both $CO_2$ and $CH_4$. The biogenic component plays a pivotal role in the variations of $\mathrm{XCO_2}$. The impact from anthropogenic emission sources is somewhat weak compared to this, while the $\mathrm{XCH_4}$ enhancement is dominated by human activities.

We then concentrated on using DCM for focusing our analysis on relevant $CO_2$ and $CH_4$ contributions from the urban area. DCM highlights that the enhancement of XCO2 over background within the inner Berlin urban area is mostly caused by anthropogenic activities. In DCM, wind direction plays a vital role to define the upwind and downwind sites, which directly influence the calculation of differential column concentrations. In the $CO_2$ tracer analysis, it turns out that $\Delta\overline{\mathrm{XCO_2}}$, the difference with respect to a mean value instead of a specific upwind site, exhibits a more visible and clearer trend, which proves that the $CO_2$ enhancement is dominated by anthropogenic activities within the urban area. We conclude that DCM, when applied with care, helps to highlight the relevant emission sources. Similarly, for $\mathrm{XCH_4}$, DCM eliminates the bias of the simulated values. Furthermore, when $\Delta\mathrm{XCH_4}$ values suffer from inconsistent wind directions, we consider $\Delta\overline{\mathrm{XCH_4}}$ to be a useful quantity for analysis.

An analysis of $\mathrm{XCO_2}$ in the Paris hot-spot region was carried out by Vogel et al. (2019). Some of their results can be compared to the conclusions we drew in this paper. In Vogel et al. (2019), the modelled $\mathrm{XCO_2}$ was calculated based on the chemistry transport model CHIMERE (2 km) and flux framework CAMS (15 km), with hourly anthropogenic emissions from the IER

and EDGAR emission inventories, and the natural fluxes prescribed by the C-Tessel model (Sect.2 in Vogel et al. (2019)). When comparing results from our simulation, the diurnal variation in the $XCO_2$ enhancement over background (Sect.3.4 and Fig.4(a) of our paper) is comparable to the findings of Vogel et al. (2019). For the analysis on the comparison of $\Delta XCH_4$ between simulations and measurements in Sect.4.1, we found that negative column concentration differences between down-

and upwind sites appear for some periods, owing to the variation of wind directions that causes the conversion of up- and downwind sites, which was also mentioned for the $\Delta XCO_2$ analysis in Vogel et al. (2019). Based on the CHIMERE-CAMS modelling framework, they showed that the strong decrease in $XCO_2$ during daytime can be linked to net ecosystem exchange, while a significant enhancement compared to the background is caused by $XCO_2$ from fossil fuel emissions, but this is often compensated by net ecosystem exchange. We utilized DCM to bring out the role of anthropogenic activities within urban areas

(see the $XCO_2$ tracer analysis in Sect.4 of our paper).

We conclude that WRF-GHG is a suitable model for precise GHG transport analysis in urban areas, especially when combined with DCM. DCM is not only useful for the direct evaluation of measurements, but also helps us to understand the results of tracer transport models, canceling out the bias caused, e.g., by initialization conditions, and highlighting regional emission sources. This case is a fundamental study for the WRF-GHG mesoscale modelling framework. Emission flux estimations using

WRF-GHG would be our further target to be demonstrated for the case of Munich. This Munich case is combined with the first worldwide permanent column measurement network designed in Munich. Various emission tracers will be run for this case in which more emission tracers (e.g., biogenic emissions from wetland for $XCH_4$, traffic emission and strong point sources' emissions in urban areas) are being separated and analyzed using the longer time period of available measurements.

In future work, we suggest running WRF-GHG for more urban areas, such that, e.g., different transport, more emission

tracers, topography, emission scenarios and the quantification of model errors can be studied. The influence from the line of the sun sight should be taken into account and the relative sensitivity analysis is suggested. The WRF-GHG mesoscale simulation framework may also be combined with microscale atmospheric transport models to simulate crucial details of emission sources and transport patterns precisely, with the aim of tracing urban GHG emissions. A further promising direction for future studies may be the application of DCM and model-based analysis to satellite measurements, to assess gradients

across column concentrations with a dense spatial sampling.

*Author contributions.*

Xinxu Zhao performed the simulations, with the support and guidance of Julia Marshall, Christoph Gerbig, Jia Chen and Stephan Hachinger. Julia Marshall provided the CAMS fields for the initialization. Jia Chen supplied anthropogenic emission source and Christoph Gerbig offered the VPRM used for the simulations. Matthias Frey and Frank Hase provided the measure-

ment data for Berlin in 2014 and fruitful discussions related to the measurements. Stephan Hachinger provided the guidance related to the running of the simulations in the Linux Cluster. Xinxu Zhao, Jia Chen and Stephan Hachinger designed the computational framework. Xinxu Zhao and Jia Chen performed the analysis of the results. Xinxu Zhao wrote the manuscript with input from all authors. All authors provided critical feedback and helped shape the research, analysis and manuscript.

*Data availability.*

The simulation data that support the findings of this study are available on request from the corresponding author. The measurement data are available at doi:10.5194/amt-8-3059-2015 (Hase et al., 2015).

*Competing interests.*

The authors declare that they have no conflict of interest.

*Acknowledgements.* We thank the personal contribution from Dr. Michal Galkowski from Max Plank Institute for Biogeochemistry for the
biogenic-related $CH_4$ flux estimates. The priori concentration profiles from the whole Atmoshpere Community Climate Model (WACCM) were provided by J.Hannigan (NCAR). Jia Chen is partly supported by Technische Universität München – Institute for Advanced Study, funded by the German Excellence Initiative and the European Union Seventh Framework Programme under grant agreement no. 291763. The simulations presented in this work have been run on the Linux Cluster (CooLMUC-2) of the Leibniz Supercomputing Centre (LRZ, Garching).

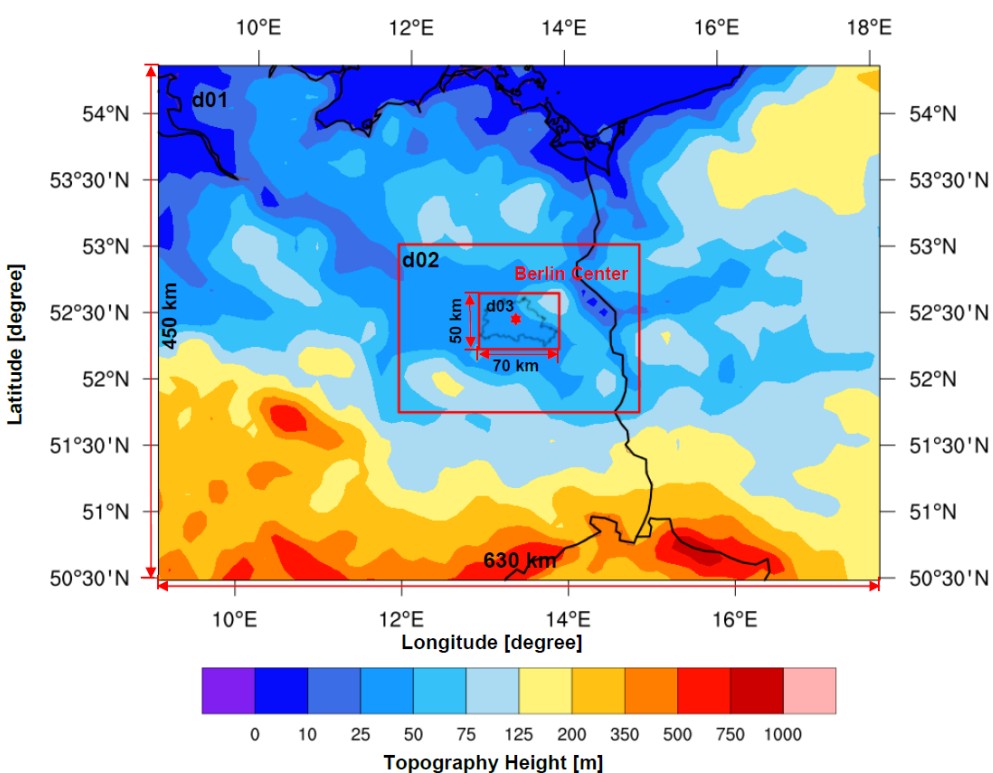

**Figure 1.** The topography map for the three domains in our study. The domain d03 is centered over Berlin, at 13.383°N, 52.517°E, marked with a red star. The boundary of Berlin from GADM (available at https://gadm.org/) is depicted in the innermost domain.

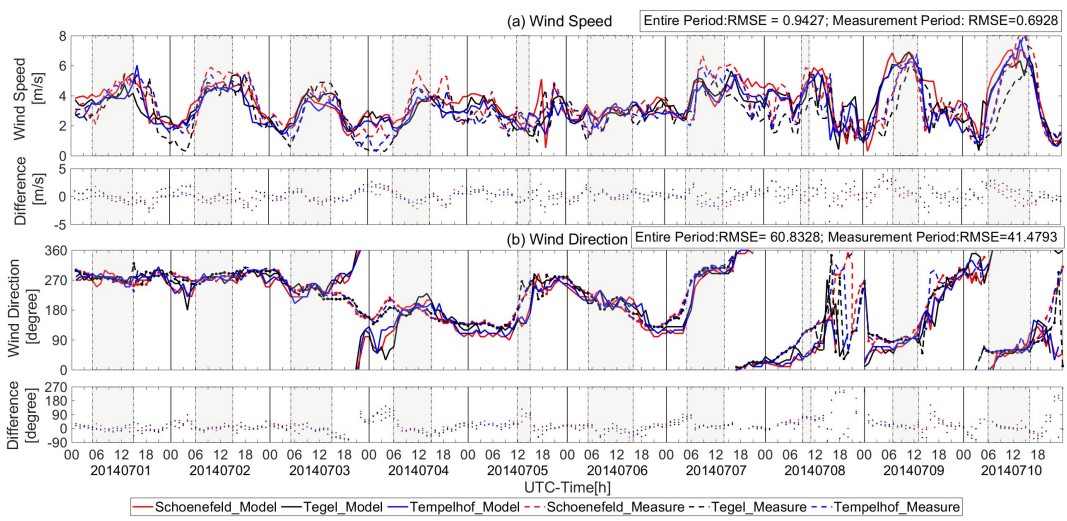

**Figure 2.** Variation and differences between simulated and measured wind fields for (a) wind speeds and (b) wind directions from 1[st] to 10[th] July 2014 at the three measurement sites, Schönefeld (red lines), Tegel (black) and Tempelhof (blue) in Berlin. The solid lines represent the simulated wind fields provided by WRF-GHG and the dashed lines depict the measured wind fields. The differences in (a)&(b) are simulations minus measurements. FTS measurement time periods on each date are marked by gray shaded areas.

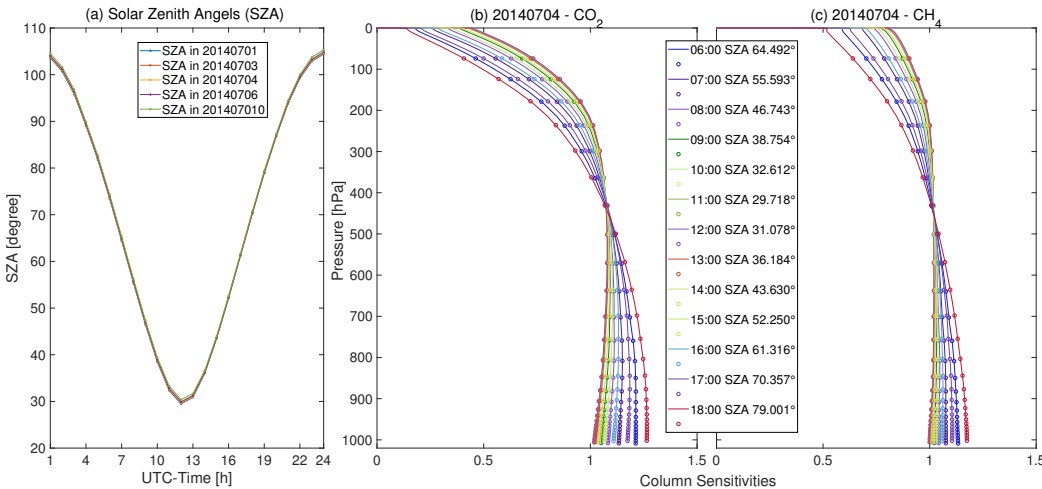

**Figure 3.** (a) Daily variations of solar zenith angle (SZA) for five simulation dates (1[st], 3[rd], 4[th], 6[th] and 10[th] July) and the vertical distributions of column sensitivities for (b) $CO_2$ and (c) $CH_4$ on 4[th] July. (b) & (c): the solid lines represent our derived column sensitivities for EM27/SUN under different SZAs, and the circles stand for the values on model pressure levels.

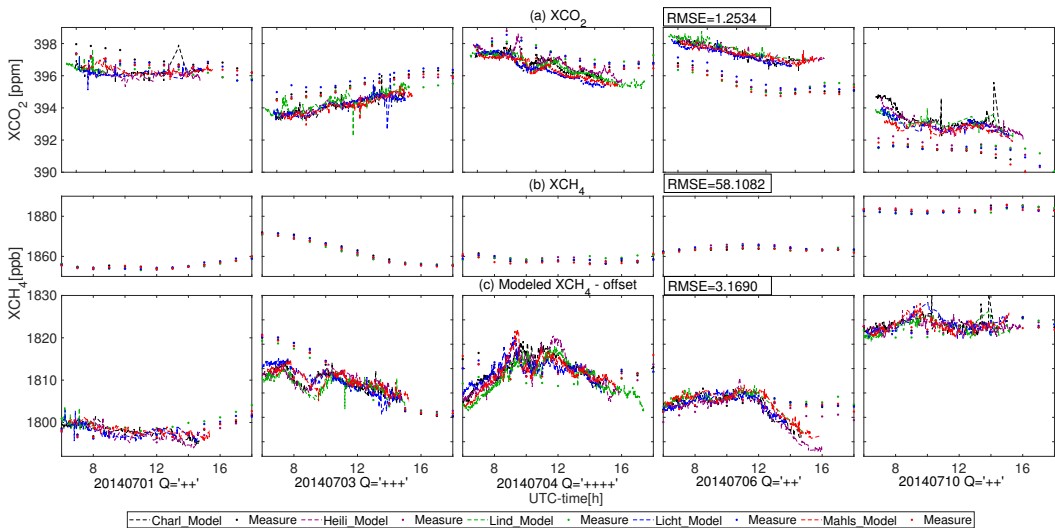

**Figure 4.** Variations of the measured and smoothed simulated (a) $XCO_2$ and (b) $XCH_4$ on 1st, 3rd, 4th, 6th and 10th July 2014, for five sampling sites in Berlin: Charlottenburg (Charl: black markers), Heiligensee (Heili: purple), Lichtenrade (Licht: green), Lindenberg (Lind: blue) and Mahlsdorf (Mahls: red). The solid circles in (a) and (b) stand for the simulated values provided by WRF-GHG and the dashed lines represent the measured concentrations. The solid circles represent the simulated $XCH_4$ after the subtraction of the daily offset in (c).

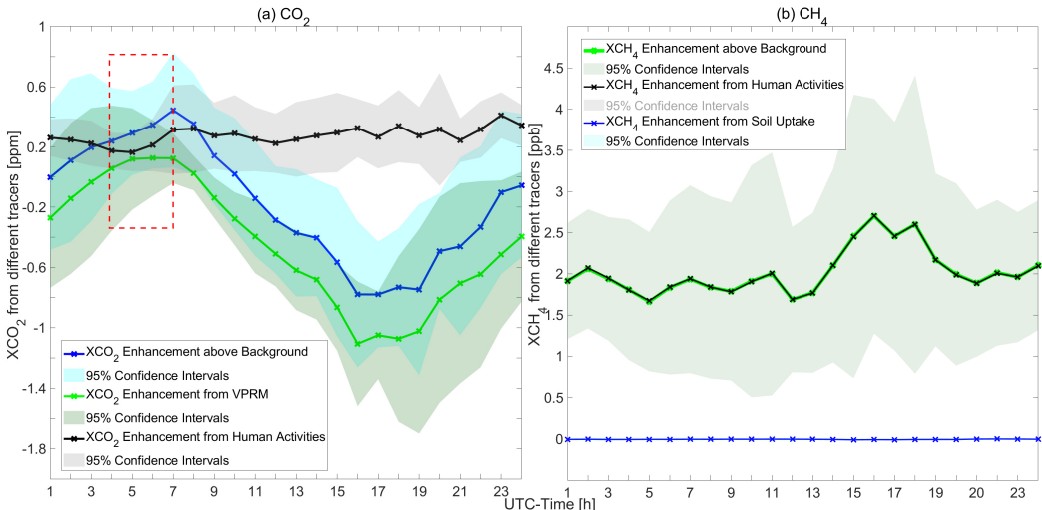

**Figure 5.** The diurnal variations of the simulated changes in concentrations caused by different emission tracers in Charlottenburg in Berlin from 2014, averaged over a period of nine days (from 2nd to 10th July 2014). The colored lines represent the concentration changes and the mean enhancement over background. (a): the mean hourly $XCO_{2,VPRM}$ (green line) and $XCO_{2,anthro}$ (black); (b): the mean hourly $XCH_{4,anthro}$ (black) and $XCH_{4,soil}$ (blue). The red box in (a) marks the morning peak of the $XCO_2$ enhancement over the background, as described in Sect.3.4.

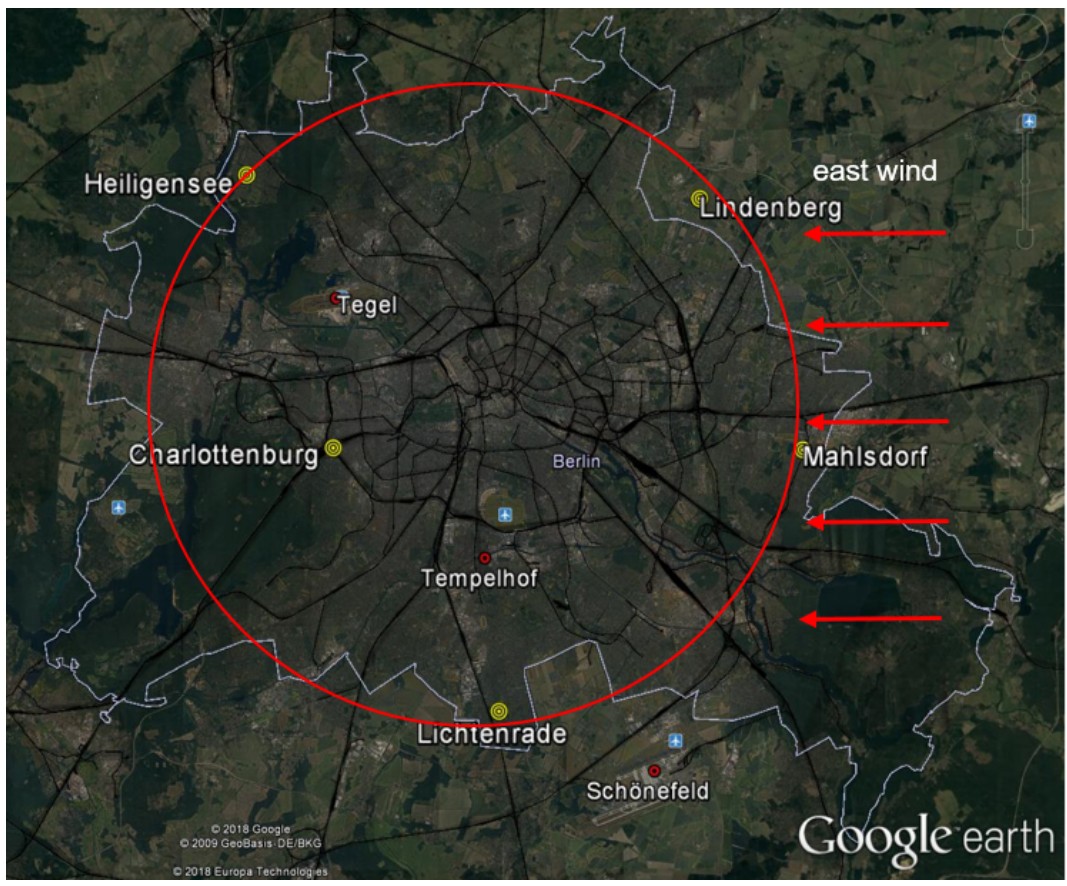

**Figure 6.** Detailed locations of the five sampling sites.The five red stars stand for the five sampling sites, four of which (Mahlsdorf, Heiligensee, Lindenberg and Lichtenrade) were roughly situated along a circle with a radius of 12 km around the center of Berlin, marked as the black circle. The innermost domain of our WRF-GHG model contains all five measurement sites. The three wind measurement sites are marked by red circles. Map provided by Google Earth.

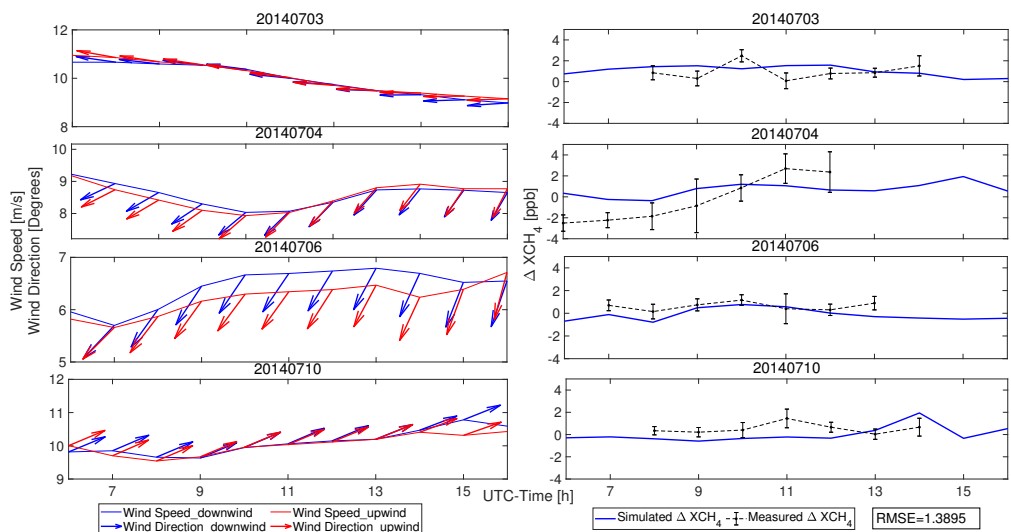

**Figure 7.** Modeled wind fields for downwind (blue lines) and upwind (red) sites (left column), and downwind-minus-upwind differential evaluation for measured (blue) and simulated (black) XCH$_4$ (right column) on 3$^{rd}$, 4$^{th}$, 6$^{th}$ and 10$^{th}$ July 2014. Based on the selection of downwind and upwind sites in Table.1, $\Delta$XCH$_4$ is calculated using Eq.6, 7 and 8, depicted by blue lines for measurements and black lines for simulations. The black error bars in the right column are the standard derivations of the minute values of the hourly mean.

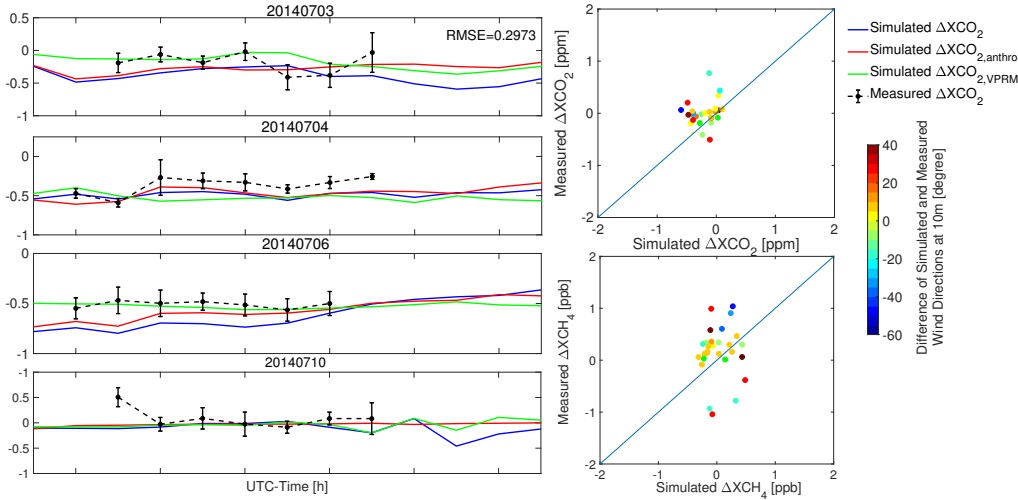

**Figure 8.** Measured (black lines) and simulated (blue) $\Delta$XCO$_2$ on 3$^{rd}$, 4$^{th}$, 6$^{th}$ and 10$^{th}$ July 2014, and Comparison of hourly mean $\Delta$XCO$_2$ and $\Delta$XCH$_4$ for these four days. The $\Delta$XCO$_2$, calculated using Eq.6, 7 and 8, are depicted by blue lines in the right column. The red and green lines show the variation of the differences between downwind and upwind sites in XCO$_2$ changes from anthropogenic and biogenic activities, respectively. The points color in right column are coded by the difference of the simulated and measured wind directions at 10 meters. The black error bars in the left column are the standard derivations of the minute values of the hourly mean.

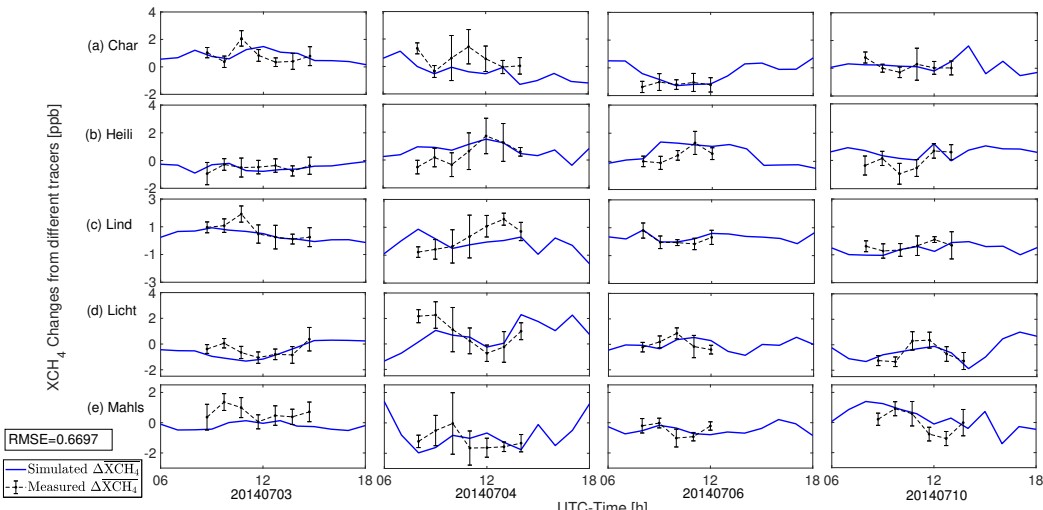

**Figure 9.** Modelled (blue lines) and observed (black) site XCH4 vs. site-mean XCH4 data for five sampling sites: Charlottenburg (Char: 1st row), Heiligensee (Heili: 2nd row), Lindenberg (Lind: 3rd row), Lichtenrade (Licht: 4th row) and Mahlsdorf (Mahls: 5th row). The black error bars in each subplot are the standard derivations of the minute values of the hourly mean.

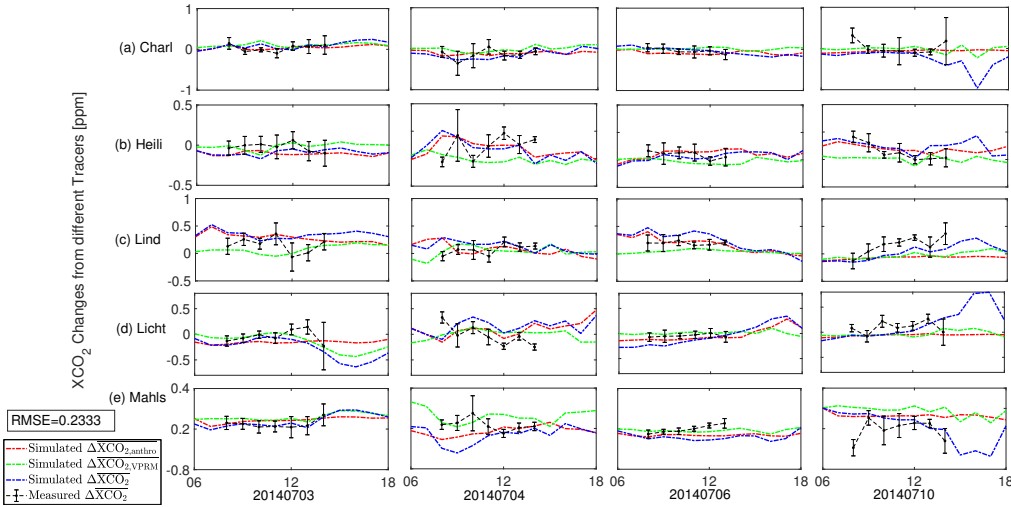

**Figure 10.** $\Delta\overline{XCO_2}$ (blue lines for simulations and black for measurements) for five sampling sites (i.e. the difference between $XCO_2$ at the site and the mean $XCO_2$ of five sampling sites): Charlottenburg (Char: 1st row), Heiligensee (Heili: 2nd row), Lindenberg (Lind: 3rd row), Lichtenrade (Licht: 4th row) and Mahlsdorf (Mahls: 5th row). We furthermore show the differences in the simulated $XCO_2$ changes from biogenic (green) and anthropogenic (red) activities. The black error bars in each subplot are the standard derivations of the minute values of the hourly mean.

**Table 1.** The selections of upwind and downwind sites for four dates

| Date | Wind Direction (degree) | Upwind Sites | Downwind Sites |
|------|------------------------|--------------|----------------|
| 3rd Jul: | 272.55 ±20.19 | Charlottenburg/Lichtenrade | Lindenberg/Mahlsdorf |
| 4th Jul: | 206.93 ±24.23 | Lichtenrade | Heiligensee/Lindenberg |
| 6th Jul: | 214.51 ±26.38 | Lichtenrade | Heiligensee/Lindenberg |
| 10th Jul: | 38.03 ±25.33 | Mahlsdorf/Lindenberg | Lichtenrade/Charlottenburg |

Wind directions are the mean of the hourly vertical-mean wind directions for one day.


## Appendix A: WRF-GHG running process

A detailed description on how to run WRF-GHG is provided in Beck et al. (2011), and thus, only the initialization process for our study in particular is summarized here. One daily simulation with WRF-GHG is normally performed for a 30-hour time period including a 6-hour spin up for the meteorology from 18 UTC to 24 UTC of the previous day and a 24-hour simulation of the tracer transport on the actual simulation day (Beck et al., 2011).

As for the boundary conditions, a small constant offset needs to be added into the WRF boundary files for the biospheric $CO_2$ and the soil sink $CH_4$ tracers at the start of each run, because these tracers can result in a net sink. When the concentrations become negative, the advected tracer fields will "disappear", as the WRF code does not allow tracers with negative values. An offset applied in the initialization process helps to avoid this problem and later is subtracted in the post-processing. As for the initial conditions, the meteorological conditions are initialized with external data sources (GFS in our model) each day to update the WRF meteorological fields properly. The tracers for the total and background $CO_2$ and $CH_4$ flux fields are initialized only once, at the first day of the simulation period, using CAMS as an external data source. Furthermore, the lateral boundary conditions of the outer domain d01 is also initialized by the CAMS. Then, for the other days within the simulation period, these tracers for the total and background $CO_2$ and $CH_4$ fluxes are directly taken from the final WRF output at 24 UTC of the previous day to make the entire simulation continuous. The $CO_2$ tracer for VPRM and the $CH_4$ tracer for soil uptake are also initialized with a constant offset to avoid the appearance of negative values caused, e.g., by the vegetation respiration (Beck et al., 2011). In terms of the other flux tracers, the tracer variables are initialized each day, using external data sources to provide the updated emission data for each tracer.

## Appendix B: Model systematic equation errors for Eq.1

In the passive tracer transport simulation, the total concentration of each GHG is represented as a separate tracer, giving redundant information (with respect to the sum of all tracers for each GHG), allowing for consistency checks. A variety of flux models and emission inventories implemented in the modules of WRF-GHG are used for the estimation of GHG fluxes. The flux values from external emission inventories are gridded and ingested into the model. In the transport process, the relationship among the changes in concentrations from different emission tracers, the total and background concentrations (Eq.1) should then be satisfied; ideally with $\Delta CO_2$ and $\Delta CH_4$ computational errors during the simulation process being zero. Nonzero values of $\Delta CO_2$ and $\Delta CH_4$ reflect the limited precision of the tracer transport calculation in WRF-GHG.

Figure.B1 thus shows the mean values (solid lines) and the 95 % confidence intervals of $\Delta CO_2$ and $\Delta CH_4$. As depicted in the figure, $\Delta CO_2$ ranges from -0.005 ppm to 0.01 ppm while $\Delta CH_4$ is in range of -0.01 ppb to 0.02 ppb. Divided by typical absolute values of the concentrations from different flux processes for $XCO_2$ (around 1 ppm) and $XCH_4$ (around 2-3 ppb) depicted in Fig.4, the relative computational error is found to be ~1 % for both $CO_2$ and $CH_4$.

These tiny computational errors can be caused by the slight non-linearity of the advection scheme used in the WRF-GHG model, which makes the sum of the concentrations in $CO_2$ and $CH_4$ from all individual flux tracers not exactly equal to the concentration from the "sum tracer", representing the total sum of all fluxes related to different processes.

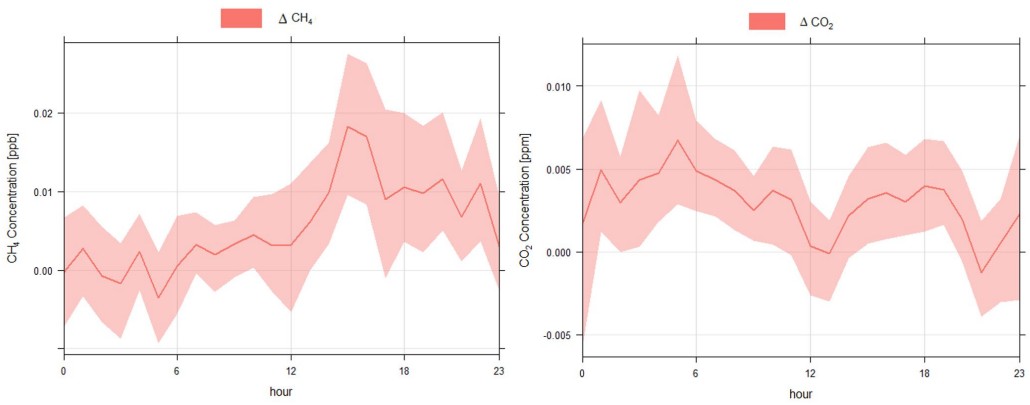

**Figure B1.** The mean values (solid lines) and the 95 % confidence intervals of the computational error $\Delta CO_2$ (left) and $\Delta CH_4$ (right). $\Delta CO_2$ and $\Delta CH_4$ are calculated using Eq.1.

## Appendix C: The vertical distribution of CH₄ in CAMS

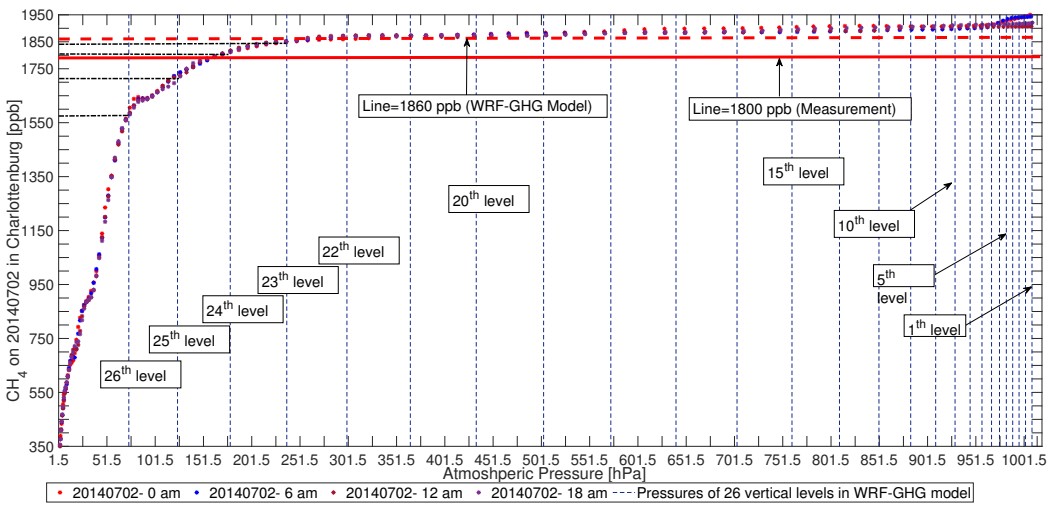

**Figure C1.** The vertical distribution of $CH_4$ on $2^{nd}$ July in Charlottenburg. The asterisks represent the $XCH_4$ field from CAMS. The vertical dashed lines show the values of atmospheric pressure corresponding to the 26 vertical levels in our WRF-GHG. Y-axis levels of 1800 ppb and 1860 ppb, corresponding to the total column measurement and the modeled value, respectively, have been marked by red horizontal (solid / dashed) lines.

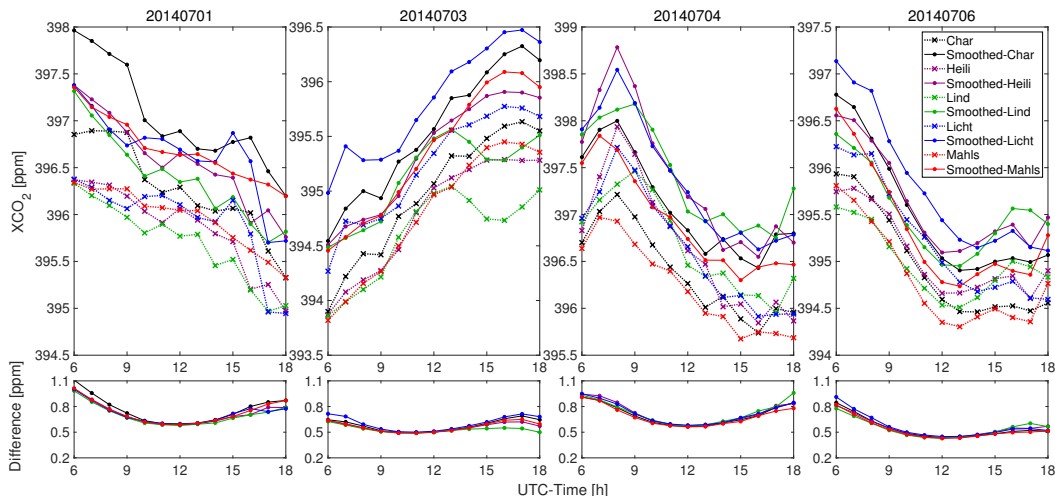

**Figure D1.** Comparison of XCO₂ from WRF-GHG with and without smoothing (using our column sensitivities for EM27/SUN) for the first four simulated dates. The five colors stand for the concentrations from five sample sites. Dotted lines with the makers 'x' represent the XCO₂ without smoothing, while solid lines with the markers 'o' stand for the smoothed values.

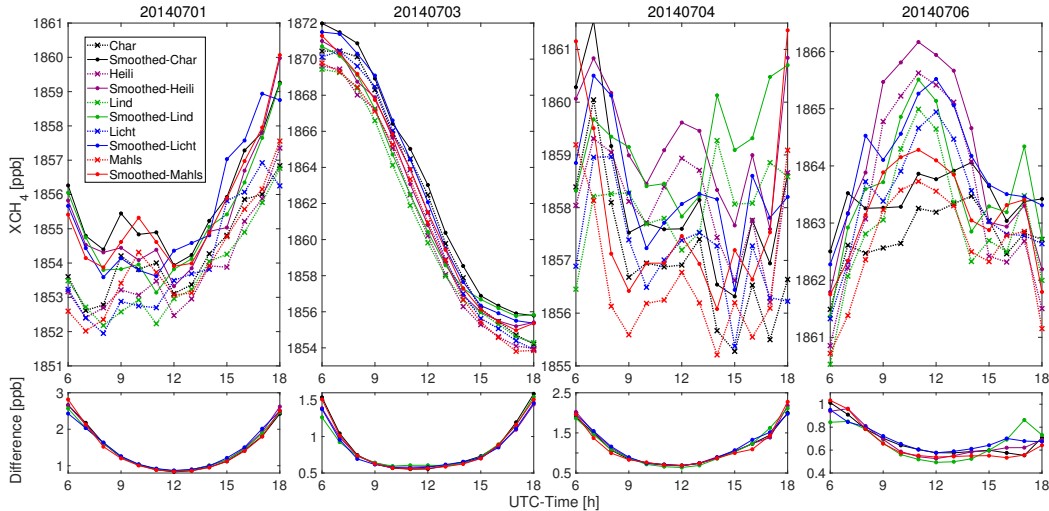

**Figure D2.** Comparison of XCH₄ from WRF-GHG with and without smoothing (using our column sensitivities for EM27/SUN) for the first four simulated dates. The five colors stand for the concentrations from five sample sites. Dotted lines with the makers 'x' represent the XCH₄ without smoothing, while solid lines with the markers 'o' stand for the smoothed values.

**490   Appendix D:  Accounting for instrumental limitations in comparison of measured to simulated XCO$_2$ and XCH$_4$**

**Appendix E:  The vertical wind profiles for wind speeds and wind directions**

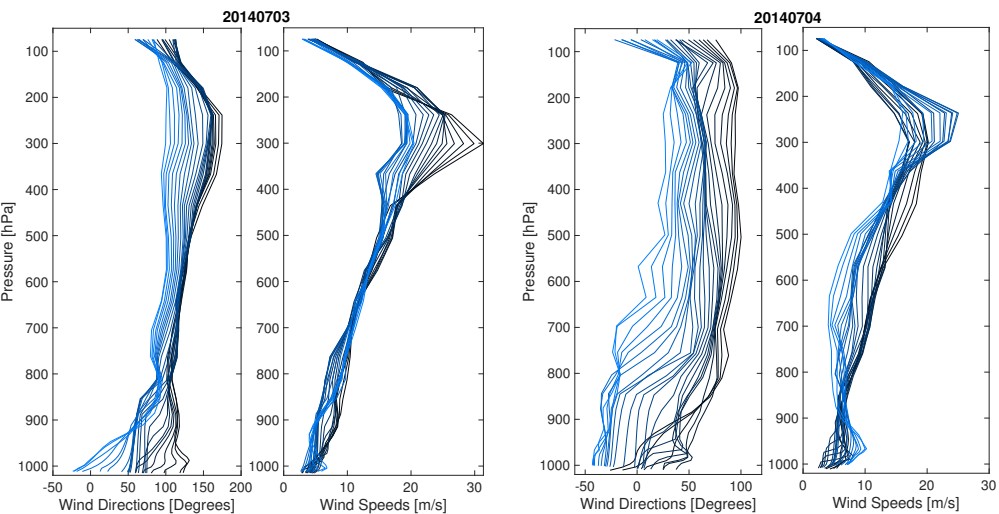

**Figure E1.** The vertical distribution of wind fields (wind speeds and wind directions) on 3$^{rd}$ July (left side) and 4$^{th}$ July (right side) in Tegel. The colors from black to blue represent the time from morning to evening.

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
