# Peer review of "Analysis of Total Column CO2 and CH4 Measurements in Berlin with WRF-GHG"

_Atmospheric Chemistry and Physics, 2018_

## Referee Comment (RC1) · Anonymous Referee #1 · 15 Feb 2019

Summary

Zhao et al. present a study on total column CO2 and CH4 in Berlin. Their results from a high-resolution modelling framework based on WRF-GHG-GFS and VPRM-EDGAR fluxes is compared to previously published observations by Hase et al. 2014. The authors found that XCO2 can be modelled reasonably well, while CH4 showed a significant bias of ca. 2.7%. Using a differential column methodology the influence of variations in the boundary conditions and non-anthropogenic sources can be reduced and the model-data mismatch of the DCM-derived XCO2 and XCH4 was further investigated.

General comments

[Figure]

Overall, the paper is well-written and the structure is straightforward. Despite only covering a few days of measurements this paper adds interesting results to a growing field of research and demonstrates the value of DCM. After addressing the general and specific comments, I would recommend this paper for publication.

The manuscript falls a bit short by not considering or discussing all potential sinks and sources of CH4, but their assumptions are largely supported by citations and other studies. Given the high temporal and spatial resolution of the modelling framework it would have also allowed to investigate other issues in more detail, e.g. which sources in the region are dominant (are the power plants the key contributors to CO2,ff in Berlin) or how would a change in the daily emission cycle affect the model-observation mismatch. The spatial and temporal disaggregation is mentioned as a major source of uncertainty in bottom-up inventories for cities, but this topic is not really discussed through the lens of the modelling and measurement results here.

Specific comments

Line 23: 50% seems to be an extreme example and it seems advisable to give the range and typical uncertainty of national emission inventory reports.

Line 37: Typical 'top-down' methods rely on prior information on fluxes, therefore, the assumption that they are 'independent' should be further explained.

Line 43: Please clarify what kind of 'carbon cycle processes' you are referring here.

Line 53: Please add information about the manufacturer for the EM27/SUN

Line 61: Vogel et al. 2018 also seem be using a very similar upwind versus downwind approach.

Line 68: Previous studies have found that urban carbon fluxes are significantly higher than predicted with conventional models like VPRM (Hardiman et al. 2017; https://doi.org/10.1016/j.scitotenv.2017.03.028). Using VPRM is thus a limitation that should be discussed.

Line 76: It would seem important to note that Berlin is actually a state, with more regulatory influence than other cities.

Line 93: Please clarify what is meant by 'actual meteorological conditions'

Line 115 – eq. 1: The equation for CH4 is missing any biogenic production of CH4. Furthermore, why is the soil sink of CH4 considered, but not the photochemical sink? Both are responsible for the CH4 lifetime.

Line 147: Were wind speeds in different heights also investigate? How could the Ekman spiral have affected your results?

Line 152+ : Why is only R2 reported? It seems the root mean square difference would be an important measure of model performance here.

Line 159: ad -> and

Line 198: Please clarify why marshy woodlands cannot be a source of biogenic CH4?

Line 210: What is meant by 'background' here? Do you consider the lowest point per day to be the background or does background refer to a theoretical Xgas value without any sources and sinks within Berlin (or a wider region)?

Line 213: 'owning' -> 'owing' – also 'wiggles' seems to be a fairly colloquial choice of words here.

Line 227: The soil sink for CH4 is significantly smaller the photo-chemical sink – why was soil uptake investigated here?

Line 248: It seems the 10m wind speed is rather a 'performance metric' or 'diagnostic parameter' rather than a 'standard'

Line 251: What were wind conditions in higher levels of the PBL?

Line 268-269: Why is only R2 given as metric of performance? Root mean squared differences could be important to investigate at all to properly judge the model-

observation mismatch.

Line 272: Why was this specific time window chosen?

Line 295: Please clarify what is meant by 'concentration fields', I would assume the instrument measures one total column Xgas value for each time step and not a higher dimensional field.

Line 310: Couldn't this also be caused by errors in the spatio-temporal distribution of emission map or other just missing non-anthropogenic sources. Furthermore, the underlying (GFS-driven) wind data is also limited in its resolution in space and time.

Line 329: Please clarify the distinction you make between being 'pivotal' and playing a leading role.

Line 331: Please clarify 'background'. If this refers to a larger scale value without the influence of sources, then this is not surprising. You ignore photo-chemistry and assume only anthropogenic CH4 sources.

Line 345: What is a 'flux framework'?

Line 346: According to the CAMS webpage their CO2 vegetation model is C-Tessel (https://atmosphere.copernicus.eu/global-production-system)

Figure 3: Symbols are hard to distinguish

Figure 4: Consider changing label to 'XCO2 enhancement from...'

Figure 6: East Wind > east wind; also please consider adding the administrative boundaries of Berlin or is all of the urban are in the plot Berlin?

Figure 10: The legend is hard to read and curve for the observations are hard to see as well. Also consider adding error bars to the observed values.

Line 460: -> Why 'assessed'?

---

## Referee Comment (RC2) · Anonymous Referee #2 · 19 Feb 2019

This paper analyzes the potential of the WRF-GHG model to simulate the column-averaged abundances of CO2 and CH4 under urban environments. The precision of such simulations is assessed by comparing to observations carried out in Berlin using portable and low-resolution EM27 spectrometers. The combination of model and observations allow the authors to highlight the potential of using the differential column methodology (DFM) to estimate urban CO2 and CH4 emissions and identify the main driver processes responsible for such emissions. The paper is well structured, but extremely concise and the results are not properly discussed in depth. I suggest this paper might be suitable for publication after general and specific comments listed below are addressed.

General comments:

[Figure]

1) The useful and potential of the differential column methodology (DFM) is investigated to evaluate the urban emissions. It is especially interesting the author's suggestion to eliminate the wind influence on this methodology. But, in general, this promising database (simulations and observations) is poorly exploited by the authors. The discussion and attribution of GHGs sources is very simple, not being supported by robust statistics methods (eg, multivariate analysis,...) and, in some parts of paper, only using one day of observations. Also, I would recommend to the authors to go in depth in this work by providing flux emission estimations from this database, similarly for example to Viatte et al. (2017). Specially, from the observation point of view, it is crucial to analyze the potential of DFM technique to provide actual GHGs flux emissions. Other interesting point would be to assess the improvements of WRF-GHG simulations if the atmospheric GHGs observations from EM27 instruments were assimilated.

2) The precision of the WRF-GHG simulations is assessed by comparing to the EM27 column observations carried out between 23rd June and 11st July 2014 in Berlin. But, the authors use different days depending on what they want to discuss without a precise explanation. For example, the wind comparison is only performed for the period from 1st July to 5th July, the comparison of column-averaged concentrations is carried out for the 3rd and 4th July. But, for the DCM analysis the authors include the 1st and 2nd July, but they rule out the 4th July for section 4.1. However, for section 4.2 the authors use 3rd, 4th, 5th, and 6th July. This limitation on dates also affects the robust of the found conclusions, since the size of the compared database is very limited (for example, the section 4.2 is almost supported by the comparison of one day, analyzing correlations with only ten points). This is very confusing. Why do not the authors use the whole field campaign observations for the wind and concentration comparison? As authors pointed out, to check the DCM approach stable conditions are needed, but these conditions must be justified and supported by the results from this work itself. Therefore, I would recommend to authors to include the whole period of the field campaign for the comparison of the wind and column-averaged concentration fields (sections 2 and 3), which will also support the identification of GHGs sources and

sinks presented in the paper. From these comparison results, the authors could clearly identify the optimal conditions to analyze the DCM approach.

Specific comments

Line 74: The novelties and obtained improvements with regards to the work of Pillai et al. (2016) should be discussed in more detail in the paper.

Line 125: One of the novelties of this work is to provide GHGs high-resolution simulations (1kmx1km). In this sense, it would be nice to discuss the impact of using, for example the EDGAR V4.1 inventory as anthropogenic fluxes at a spatial resolution of $0.1°$ (about 10 kmx10 km), on the simulated concentrations fields, or how the spatial resolution is treated in the VPRM model.

Line 129: The authors mention that the time factors for the assimilated anthropogenic fluxes could introduce considerable uncertainties. Please, explain more in detail this issue, eg, the value of these uncertainties or how these could affect the GHGs concentration simulations.

Line 152. The field campaign is carried out between 23rd June and 11st July 2014. Why is the period from 1st July to 5th July only used for the wind comparison? Since EM27 observations are performed during day-time, it would be interesting to include the performance analysis for wind comparison distinguishing between day and night-time.

Line 159. Explain possible reasons for the discrepancies between the model and observed wind fields. Maybe the model is not able to capture very fast changes of air masses, which will strongly affect the GHG comparisons.

Line 166. The influence of the EM27 vertical sensitivity is neglected in the model-observation comparison. But, the EM27 averaging kernels show a dependence on the solar geometry, which is considered in similar studies such as Vogel et al. (2018). Please justify more in detail why not to smooth the GHGs simulations (using the EM27

averaging kernels and a priori profile).

Line 176. For the comparison between simulated and observed column-averaged concentrations, the authors limit the analysis to the 3rd and 4th July. But, according to the wind comparison, the 3rd July shows the highest discrepancies between model and simulations. Therefore, observations and simulations could account for different air masses with different GHGs concentrations and influenced by different sources, which should be considered in the discussion. To account for this, the authors could plot the simulated-observed differences as a function of the wind differences. Although the EM27 stations do not coincide with the meteorological stations, figure 2 shows that the study area is very homogenous with regards to wind fields. Thereby, these differences could be a proxy of the model inconsistences.

Adjust better the scales of figure 3 to clearer see the dispersion and distribution of the data, especially for CH4. Now, it is hard to identify what database shows more variability, which could be interesting to know if the simulations are underestimating or overestimating the real CH4 variability.

Line 187-194. Although applying the offset seems to improve the comparison between observations and simulations, the model is not capturing well the observed CH4 variability (R2 is too small), thereby model and observations are not reflecting the same air masses, sources (industries or natural processes). The authors superficially mention the possible influence of the tropopause height in the simulations, but without quantifying this impact. Have the authors considered the possible influence of the PBL? Or the shape of the constant a priori profile used for EM27 retrievals? Please include a more detailed discussion of the possible reasons for these discrepancies.

Line 195-235: As mentioned in the general comments, the discussion and attribution of GHGs sources is very simple. Please consider to include a robust statistics analysis to support the main conclusions of this section.

Line 248. Have the authors analyzed the vertical distribution of the winds within PBL

for the comparisons?

Line 260-289. A plot showing the CH4 and CO2 enhancement observed and simulated as a function of the wind directions or differences between simulated and observed wind directions could help to explain better the results of the section 4. Regarding to Figure 7 and 8, why are not the wind fields considered in Figure 7 similarly to Figure 8? Why does not Figure 8 include the 1st July? Why are not the performance values for CO2 included in the text?
* * *

---

## Author Comment (AC1) · 15 May 2019

Zhao et al. present a study on total column $CO_2$ and $CH_4$ in Berlin. Their results from a high-resolution modelling framework based on WRF-GHG-GFS and VPRM-EDGAR fluxes is compared to previously published observations by Hase et al.2014. The authors found that $XCO_2$ can be modelled reasonably well, while $CH_4$ showed a significant bias of ca. 2.7%. Using a differential column methodology the influence of variations in the boundary conditions and non-anthropogenic sources can be reduced and the model-data mismatch of the DCM-derived $XCO_2$ and $XCH_4$ was further investigated.

We thank the anonymous Referee #1 for their time and valuable comments to improve this manuscript. The general and specific comments are addressed by point-by-point detailed replies below. Referee's comments have been repeated in black. Author's replies and edited contents are marked in blue and red, respectively. We would like to add Frank Hase and Matthias Frey as the co-authors in this paper because of their contributions in the measurements and in clarifying our analysis during the review phase.

**General comments**

Overall, the paper is well-written and the structure is straightforward. Despite only covering a few days of measurements this paper adds interesting results to a growing field of research and demonstrates the value of DCM. After addressing the general and specific comments, I would recommend this paper for publication.

We thank the anonymous referee #1 for the careful reading of the manuscript and helpful comments. We have addressed the specific and general comments.

The manuscript falls a bit short by not considering or discussing all potential sinks and sources of $CH_4$, but their assumptions are largely supported by citations and other studies. Given the high temporal and spatial resolution of the modelling framework it would have also allowed to investigate other issues in more detail, e.g. which sources in the region are dominant (are the power plants the key contributors to $CO_2$,ff in Berlin) or how would a change in the daily emission cycle affect the model-observation mismatch. The spatial and temporal dis-aggregation is mentioned as a major source of uncertainty in bottom-up inventories for cities, but this topic is not really discussed through the lens of the modelling and measurement results here.

In this study, we do not cover all the processes related to the tracers and mainly focus on the dominant emission tracers for $CO_2$ and $CH_4$ within urban areas. In the 'online' tracer calculation for anthropogenic emissions, the $CO_2$ and $CH_4$ surface fluxes are derived from the EDGAR V4.1 emission inventory, and mixed vertically and horizontally based on the meteorological field. We make the assumption that the attributions from EDGAR V4.1 are reliable in our study. In view of the proportions of different emission sectors in EDGAR V4.1, energy industry and road transportation are the key contributors to $CO_2$ anthropogenic emissions in Berlin while extraction and distribution of fossil fuels are the largest $CH_4$ emitters. For the spatial and temporal dis-aggregation, the coarser emission inventory could ignore the key emission points potentially (Line 139-149), and the meteorological data might not be able to capture the entire transport features of air masses (Line 182-185).

**Specific comments**

Line 23: 50% seems to be an extreme example and it seems advisable to give the range and typical uncertainty of national emission inventory reports.

Response: We thank the referee for pointint it out. Indeed, 50% is a specific value. The range of the uncertainty on individual regional and national total fossil-fuel carbon dioxide emissions is from a few percent to more than 50% (Andres et al., 2012). We have re-phrased the sentence as follows (Line 32-34):

This approach has some uncertainty, e.g., on the national fossil-fuel $CO_2$ emission estimates, ranging from a few percent (e.g., 3%-5% for the US) to a maximum of over 50% for countries with less resources for data collection and poor

statistical framework (Andres et al., 2012).

Line 37: Typical 'top-down' methods rely on prior information on fluxes, therefore, the assumption that they are 'independent' should be further explained.

Response: Thanks for pointing out this unclear description. Estimates from the 'top-down' approach on local to global scales do rely on bottom-up estimates as priors. Advanced measurement technologies and high-resolution models are able to estimate regional emissions on the basic of the 'top-down' approach such that regional or national bottom-up emission inventories can be assessed and verified (Wunch et al., 2009; Montzka et al., 2011; Bergamaschi et al., 2018). Thus, the 'top-down' approach can help to identify the discrepancy compared to the 'bottom-up' approach and highlight the uncertainties in both methods. We have re-phrased it in Lines 37-38 as follows:

The 'top-down' approach can not only provide estimated global fluxes, but also verify the consistency and assess the uncertainties of bottom-up emission inventories (Wunch et al., 2009; Montzka et al., 2011; Bergamaschi et al., 2018).

Line 43: Please clarify what kind of 'carbon cycle processes' you are referring here.

Response: The phrase 'Carbon cycle process' describes the process in which carbon, released from emission sources, travels from the atmosphere to organisms and the earth and then back into the atmosphere, i.e., carbon sources and sinks. We have clarified in Line 42-45:

Such measurements, i.e. measurements of concentration averaged over a column of air, are performed to help to disentangle the effects of atmospheric mixing from the surface exchange (Wunch et al., 2011) and decrease the biases associated with estimates of carbon sources and sinks in atmospheric inversions (Olsen and Randerson, 2004).

Line 53: Please add information about the manufacturer for the EM27/SUN

Response: We have included the manufacturer for the EM27/SUN in Line 53-54:

Chen et al.(2016) applied the DCM using compact Fourier Transform Spectrometers (FTS) EM27/SUN (Bruker Optik, Germany).

Line 61: Vogel et al. 2018 also seem be using a very similar upwind versus downwind approach.

Response: Vogel et al. (2018) applied a similar upwind-downwind approach to determine the major contribution of $CO_2$ in urban area using the case of Paris. In their station-to-station calculations, one site (RES) is taken as the upwind reference to assess the impact of local sources (details in Section 3.2.3 of Vogel's paper), whereas we considered the wind direction when selecting the downwind and upwind sites in our case. The details are discussed in the conclusion (Line 374-390) of the content.

Line 68: Previous studies have found that urban carbon fluxes are significantly higher than predicted with conventional models like VPRM (Hardiman et al.2017: https://doi.org/10.1016/j.scitotenv.2017.03.028). Using VPRM is thus a limitation that should be discussed.

Response: Thank you for your suggestions. We discussed this line in Lines 72-75 as follows:

Biogenic carbon fluxes given by the VPRM model tend to underestimate urban ecosystem carbon exchange, owing to the incomplete understanding of urban vegetation, and to conditions related to urban heat islands and altered urban phenology (Hardiman et al., 2017).

Line 76: It would seem important to note that Berlin is actually a state, with more regulatory influence than other cities.

Response: Thanks for pointing it out. We have included the regulatory influence in Line 86-88, as follows:

With its strong regulatory influence as a 'state' within Germany, and a strongly supportive policy, Berlin has already transformed itself into a climate-friendly city in which $CO_2$ emissions have been reduced by a third compared with 1990 levels, aiming for carbon neutrality by 2050 (Homann, 2018).

Line 93: Please clarify what is meant by 'actual meteorological conditions'.

Response: The actual meteorological conditions refers to a model initialization using real data for meteorological fields in the pre-processing, instead of idealized initialization. We have clarified in Line 104-106:

...based on the actual meteorological data in this case. The meteorological initial conditions and lateral boundary conditions were taken from the Global Forecast System (GFS) model reanalysis in which in-situ measurements and satellite observations have been assimilated.

Line 115 – eq. 1: The equation for $CH_4$ is missing any biogenic production of $CH_4$. Furthermore, why is the soil sink of $CH_4$ considered, but not the photochemical sink? Both are responsible for the $CH_4$ lifetime.

Line 227: The soil sink for $CH_4$ is significantly smaller the photo-chemical sink – why was soil uptake investigated here?

Line 198: Please clarify why marshy woodlands cannot be a source of biogenic $CH_4$?

Response: Thanks for these valuable comments about the $CH_4$ tracer analysis. We do agree that with a $CH_4$ lifetime of approx. 9 years, neither soil uptake nor photo-chemical sink would have a strong impact in our short simulation period (10 days), mentioned in Line 153-155 of the content.

The main reason why the soil-uptake process is taken into consideration in this study is that Berlin is located in an area of low-lying, marshy woodlands. We doubted that marshy woodlands soils might be potentially effective $CH_4$ sinks and can be quantified in the soil uptake model which was already built in WRF-GHG by Dr. Veronika Beck. As seen clearly in Fig.5 of the content, we concluded that the emissions from the soil-uptake process have almost no influence on the daily variations of $XCH_4$.

This soil uptake model built in WRF-GHG is a process-based model for the calculation of the consumption of atmospheric $CH_4$ by soils (the result of an entirely biological oxidation process, including the diffusion and microbial oxidation processes). Generally, through simplifying the physiology of the methanotrophs, this process-based model calculates the $CH_4$ fluxes into soil based on the activity of methanotrophs (i.e., the potential rate of $CH_4$ oxidation within the soil), a number environmental factors (soil temperature, soil type, moisture, etc..), the diffusivity of the topsoil, the first-order oxidation rate, etc.. The details can be found in Beck et al., 2011.

For the photo-chemistry in the troposphere, in turn, the chemical reaction of 'OH' production in the lower stratosphere and upper troposphere needs to be quantified. At the moment, the WRF-GHG model allows for passive tracer transport simulations, i.e. without any chemical reactions of $CH_4$ mentioned in Sect.2. In further studies we may consider to add an additional 'off-line' flux chemical model or the corresponding process-based model to quantify the changes in $CH_4$ concentrations from photochemistry and combine the estimates into our simulation, mentioned in the conclusion of the content.

Woodland soils can be definitely regarded as an effective $CH_4$ source. During the process of methanogenesis, methane is a byproduct in hypoxic conditions, which are common in wetlands, where they are responsible for marsh gas. We thank Dr. Michal Galkowski from Max Plank Institute for Biogeochemistry who provided us with the biogenic-related $CH_4$ emissions for the area closer to the city Berlin region (242 km × 202 km) based on simulations for 2018 using the updated version of WRF-GHG. In Dr.Galkowski's study, wetlands contributes around 15% of the total $CH_4$ emissions (including anthropogenic, dispersed sources and point sources, wetland and termites) for his domain. Wetlands sources in this domain are mainly attributed to the area in the southern part which is roughly near Biosphärenreservat Spreewald (approx. 60 km away from the Berlin center) and is not included in the innermost domain of our study. Meanwhile, as described in our content (Sect 3.4), there is no wetland in the city of Berlin according to the MODIS Land Cover Map. Thus, we can have an estimate about the magnitude of biogenic $CH_4$ emissions based on Dr.Galkowski's data and find that the influence for $CH_4$ biogenic emissions from wetlands is quite weak in our innermost domain. In this study, our interest is mainly on the major emission tracers. So we did not consider the wetland source as the targeted $CH_4$ emission tracer in the analysis for the city of Berlin.

Line 147: Were wind speeds in different heights also investigate? How could the Ekman spiral have affected your results?

Response: We did a brief analysis on the vertical distribution of wind fields. As seen from Fig.1, above approx.300 hPa, the lower the pressure is, the larger the wind speed is. While the wind speed decrease sharply with the increase of the

[Figure]

Figure 1: The vertical distribution of wind fields (wind speeds and wind directions) on 3$^{rd}$ July (left side) and 4$^{th}$ July (right side) in Tegel. The colors from black to blue represent the time from morning to evening.

height (below 300 hPa). Overall, wind directions shift from northeast to northwest from morning to evening. As depicted in the trend of the vertical distribution of wind directions, the surface wind shows a prevailing pattern towards the east with the increase of the altitude in the troposphere. Then above the troposphere, the prevailing pattern turns gradually to be in the opposite direction (the east). The vertical distributions of wind fields are also shown in the Appendix.E of the content.

Line 152+ : Why is only $R^2$ reported? It seems the root mean square difference would be an important measure of model performance here.
Line 268-269: Why is only $R^2$ given as metric of performance? Root mean squared differences could be important to investigate at all to properly judge the model-observation mismatch.

Response: Thanks for your valuable suggestions. We do agree with your point. Root mean square error (RMSE) is definitely a more appropriate measure to describe how accurately the model simulates and indicates the absolute value to show how simulated values are close to measure data points. Lower RMSE shows better fits generally. In our rephrased content, we have provided RMSEs in figures and RMSE also helps to evaluate the performance of two calculations in differential column method.

Line 159: ad → and

Response: Change made.

Line 210: What is meant by 'background' here? Do you consider the lowest point per day to be the background or does background refer to a theoretical Xgas value without any sources and sinks within Berlin (or a wider region)?

Response: Sorry for this unclear point. The 'background' concentrations here are Xgas values without the influence from any sources within the domain area. Correspondingly, the total concentrations are the combination between the background concentrations and the concentration changes from different tracers. The background and total concentration fields are initialized by the CAMS dataset. We have clarified this point in Line 127.

Line 213: 'owning' → 'owing' – also 'wiggles' seems to be a fairly colloquial choice of words here.

Thanks for pointing it out. We have changed these two parts in Line 286.

Line 248: It seems the 10m wind speed is rather a 'performance metric' or 'diagnostic parameter' rather than a 'standard'.

Response: We changed the standard condition for the selection of downwind and upwind sites into the simulated daily mean wind directions (see the left column of Fig.7). Details are discussed in Sect 4.1 and Table.1.

Line 251: What were wind conditions at higher levels of the PBL?

Response: As described in Sect.2, the vertical layers in WRF model follows the pressure definition and the upper pressure in our case is up to tropopause height. Figure.2 shows the wind fields within PBL on 3$^{rd}$ July. The PBL is situated either in the second or third layer (morning and night) or in the 13$^{th}$ layer (noon) in our domain. The wind speeds and wind directions closer to the PBL are generally higher than the surface wind fields.

[Figure]

Figure 2: Variations of the simulated Planetary Boundary Layer (PBL) Height (left side), and the wind speeds (middle) and wind directions (right) within PBL on 3$^{rd}$ July at Tegel. The colors from black to blue represent the time from morning to evening. The bold solid lines represent the values within PBL.

Line 272: Why was this specific time window chosen?

Response: This time window was chosen because there is a wind switch at 10 meters in the measurements which was not seen in the simulations. In the rephrased content, we used the daily mean wind directions as the reference and re-define the calculation of delta concentrations. This daily mean wind directions average the hourly vertical-mean wind directions. This specific time window is not considered any more.

Line 295: Please clarify what is meant by 'concentration fields', I would assume the instrument measures one total column Xgas value for each time step and not a higher dimensional field.

Response: Thanks for pointing it out. The sentence has been re-phrased for 'The instruments measure the concentration value every minute (Hase et al., 2015).'

Line 310: Couldn't this also be caused by errors in the spatio-temporal distribution of emission map or other just missing non-anthropogenic sources. Furthermore, the underlying (GFS-driven) wind data is also limited in its resolution in space and time.

Response: Thanks for your valuable suggestion. Indeed, these could be potentially the reasons why the simulations show stable variations compared to the measurements. The sentence in Line 250-253 has been rephrased:

The smaller variations from the simulation results can, e.g., be caused by the error from the spatial-temporal treatment of emission maps, underestimated emissions from anthropogenic activities, the coarse wind data and/or the smoothing

of actual extreme values in the simulation.

Line 329: Please clarify the distinction you make between being 'pivotal' and playing a leading role.

Response: Sorry for this confusion. Basically, the words 'pivotal' and 'leading' both are used to emphasize the importance and the large contribution of one object. To avoid this confusion, we only keep 'pivotal' in the content.

Line 331: Please clarify 'background'. If this refers to a larger scale value without the influence of sources, then this is not surprising. You ignore photo-chemistry and assume only anthropogenic $CH_4$ sources.

Response: Thanks for pointing out this unclear point. Given the very limited size of the domain, ignoring the methane photo-chemistry would have at most a very small effect on this offset. As described above, the biogenic-related $CH_4$ emissions contribute very little to the $CH_4$ emissions, while the $CH_4$ bias is around 50 ppb. Furthermore, the background, as described above, is taken from CAMS, a global atmospheric composition analysis which takes the photochemical sink into account.

Line 345: What is a 'flux framework'?

Response: The 'flux framework' stands for a suite of models in the operational CAMS global assimilation and forecasting system, which is directly referred from Vogel et al., 2019.

Line 346: According to the CAMS webpage their $CO_2$ vegetation model is C-Tessel (https://atmosphere.copernicus.eu/global-production-system)

Response: Thanks for pointing it out and it has been corrected.

Figure 3: Symbols are hard to distinguish.

Response: Thanks for your suggestion and the figure has been re-plotted.

Figure 4: Consider changing label to 'XCO2 enhancement from. . .'.

Response: The edits have been made in Fig.5.

Figure 6: East Wind > east wind; also please consider adding the administrative boundaries of Berlin or is all of the urban are in the plot Berlin?

Response: The edits have been made in the new figure.

Figure 10: The legend is hard to read and curve for the observations are hard to see as well. Also consider adding error bars to the observed values.

Response: The error bars have been added for all the measurement-related values for concentration fields and the legends are enlarged.

Line 460: $\rightarrow$ Why 'assessed'?

Sorry for this wrong word. It should be 'accessed' instead of 'assessed'. The 'accessed' is used to show the time when the data or information was taken from the link.

**References**

Andres, R. J., Boden, T. A., Bréon, F.-M., Ciais, P., Davis, S., Erickson, D., Gregg, J. S., Jacobson, A., Marland, G., Miller, J., et al.: A synthesis of carbon dioxide emissions from fossil-fuel combustion, Biogeosciences, 9, 1845–1871, https://doi.org/10.5194/bg-9-1845-2012, 2012.

Bergamaschi, P., Karstens, U., Manning, A. J., Saunois, M., Tsuruta, A., Berchet, A., Vermeulen, A. T., Arnold, T., Janssens-Maenhout, G., Hammer, S., Levin, I., Schmidt, M., Ramonet, M., Lopez, M., Lavric, J., Aalto, T., Chen, H., Feist, D. G., Gerbig, C., Haszpra, L., Hermansen, O., Manca, G., Moncrieff, J., Meinhardt, F., Necki, J., Galkowski, M., O'Doherty, S., Paramonova, N., Scheeren, H. A., Steinbacher, M., and Dlugokencky, E.: Inverse modelling of European $CH_4$ emissions during 2006–2012 using different inverse models and reassessed atmospheric observations, Atmos. Chem. Phys., 18, 901-920, https://doi.org/10.5194/acp-18-901-2018, 2018.

Hardiman, B. S., Wang, J. A., Hutyra, L. R., Gately, C. K., Getson, J. M., and Friedl, M. A.: Accounting for urban biogenic fluxes in regional carbon budgets, Science of the Total Environment, 592, 366–372, https://doi.org/10.1016/j.scitotenv.2017.03.028, 2017.

Homann, G.: Climate Protection in Berlin, Tech. rep., Senate Department for the Environment, Transport and Climate Protection, https://www.berlin.de/senuvk/klimaschutz/politik/download/klimaschutzpolitik_en.pdf, 2018.

Kirschke, S., Bousquet, P., Ciais, P., Saunois, M., Canadell, J.G., Dlugokencky, E.J., Bergamaschi, P., Bergmann, D., Blake, D.R., Bruhwiler, L. and Cameron-Smith, P., Three decades of global methane sources and sinks. Nature geoscience, 6(10), p.813, 2013.

Montzka, S. A., Dlugokencky, E. J., and Butler, J. H.: Non-$CO_2$ greenhouse gases and climate change, Nature, 476, 43–50, https://doi.org/10.1038/nature10322, 2011.

Olsen, S. C. and Randerson, J. T.: Differences between surface and column atmospheric $CO_2$ and implications for carbon cycle research, Journal of Geophysical Research: Atmospheres, 109, https://doi.org/10.1029/2003JD003968, 2004.

Ridgwell, A. J., Stewart, J. M. Keith, G.: Consumption of atmospheric methane by soils: A process-based model. Global Biogeochemical Cycles, 13.1, 59-70, https://doi.org/10.1029/1998GB900004, 1998.

Vogel, F. R., Frey, M., Staufer, J., Hase, F., Broquet, G., Xueref-Remy, I., Chevallier, F., Ciais, P., Sha, M. K., Chelin, P., Jeseck, P., Janssen, C., Té, Y., Groß, J., Blumenstock, T., Tu, Q., and Orphal, J.: $XCO_2$ in an emission hot-spot region: the COCCON Paris campaign 2015, Atmos. Chem. Phys., 19, 3271-3285, https://doi.org/10.5194/acp-19-3271-2019, 2019.

Wunch, D., Wennberg, P. O., Toon, G. C., Keppel-Aleks, G., and Yavin, Y. G.: Emissions of greenhouse gases from a North American megacity, Geophys. Res. Lett., 36, L15810, https://doi.org/10.1029/2009GL039825, 2009.

Wunch, D., Toon, G. C., Blavier, J.-F. L., Washenfelder, R. A., Notholt, J., Connor, B. J., Griffith, D. W., Sherlock, V., and Wennberg, P. O.: The total carbon column observing network, Philosophical Transactions of the Royal Society of London A: Mathematical, Physical and Engineering Sciences, 369, 2087–2112, https://doi.org/10.1098/rsta.2010.0240, 2011.

---

## Author Comment (AC3) · 15 May 2019

This paper analyzes the potential of the WRF-GHG model to simulate the column-averaged abundances of $CO_2$ and $CH_4$ under urban environments. The precision of such simulations is assessed by comparing to observations carried out in Berlin using portable and low-resolution EM27 spectrometers. The combination of model and observations allow the authors to highlight the potential of using the differential column methodology (DFM) to estimate urban $CO_2$ and $CH_4$ emissions and identify the main driver processes responsible for such emissions. The paper is well structured, but extremely concise and the results are not properly discussed in depth. I suggest this paper might be suitable for publication after general and specific comments listed below are addressed.

We thank the anonymous Referee #2 for their time and valuable comments to improve this manuscript. The general and specific comments are addressed by point-by-point detailed replies below. Referee's comments have been repeated in black. Author's replies and edited contents are marked in blue and red, respectively. We would like to add Frank Hase and Matthias Frey as the co-authors in this paper because of their contributions in the measurements.

**General comments**

1) The useful and potential of the differential column methodology (DFM) is investigated to evaluate the urban emissions. It is especially interesting the author's suggestion to eliminate the wind influence on this methodology. But, in general, this promising database (simulations and observations) is poorly exploited by the authors. The discussion and attribution of GHGs sources is very simple, not being supported by robust statistics methods (eg, multivariate analysis,...) and, in some parts of paper, only using one day of observations. Also, I would recommend to the authors to go in depth in this work by providing flux emission estimations from this database, similarly for example to Viatte et al. (2017). Specially, from the observation point of view, it is crucial to analyze the potential of DFM technique to provide actual GHGs flux emissions. Other interesting point would be to assess the improvements of WRF-GHG simulations if the atmospheric GHGs observations from EM27 instruments were assimilated

Response: Thanks for your valuable comments. The focus of this study is to evaluate the behavior of WRF-GHG in urban areas and test whether differential column methodology could potentially be a proper method in the model analysis to cancel out the bias from background concentrations and highlight the major emission tracers within limited regions. Emission flux estimations using WRF-GHG would be our further target which plans to be demonstrated in the Munich case. This Munich case is combined with the first worldwide permanent column measurement network designed in Munich. Various emission tracers are suggested to run for this case in which more emission tracers (e.g., biogenic emissions from wetland for $XCH_4$, the traffic emission and strong point sources' emissions in urban areas) are being added and the long-period data can also be available. Thus, this Berlin case is a fundamental study for the WRF-GHG model with high resolutions in urban areas and testing whether differential column methodology can work for the model analysis in urban regions. The reason why we only consider two dates for concentration comparison at the beginning is the limitation of EM27 measurement status and we chose to follow Hase et al.,(2015) and target the two best measurement dates. In the rephrased version, all the dates with the relatively good measurement qualities (above 'fair') are shown. And we also included the calculation of the smoothed simulations using the EM27 vertical sensitivity and compare the smoothed values to the values without smoothing which are described in Sec.3.3 and Appendix.D.

2) The precision of the WRF-GHG simulations is assessed by comparing to the EM27 column observations carried out between 23[rd] June and 11[th] July 2014 in Berlin. But, the authors use different days depending on what they want to discuss without a precise explanation. For example, the wind comparison is only performed for the period from 1[st] July to 5[th] July, the comparison of column-averaged concentrations is carried out for the 3[rd] and 4[th] July. But, for the DCM analysis the authors include the 1[st] and 2[nd] July, but they rule out the 4[th] July for section 4.1. However, for section 4.2 the authors use 3[rd], 4[th], 5[th], and 6[th] July. This limitation on dates also affects the robust of the found conclusions, since the size of the compared database is very limited (for example, the section 4.2 is almost supported by the comparison of one day, analyzing correlations with only ten points). This is very confusing. Why do not the authors use the whole field campaign observations for the wind and concentration comparison? As authors pointed out, to check the DCM approach stable conditions are needed, but these conditions must be justified and supported by the results from this work itself. Therefore, I would recommend to authors to include the whole period of the field campaign for the comparison of the wind and column-averaged concentration fields (sections 2 and 3), which will also support the identification of GHGs

sources and sinks presented in the paper. From these comparison results, the authors could clearly identify the optimal conditions to analyze the DCM approach.

Response: Thank you for this comment. We had shown limited (representative) time periods in order to improve clarity, but we agree with the referee that our choice may not have been optimum. So in the rephrased manuscript, we took all the dates when the measurement qualities are above 'fair' into account (basically, 5 simulation dates, 1st, 3rd, 4th, 6th and 10th July).

**Specific comments**

Line 74: The novelties and obtained improvements with regards to the work of Pillai et al. (2016) should be discussed in more detail in the paper.

Response: We have included in Line 77-85:

Pillai et al. (2016) utilized a Bayesian inversion approach based on WRF-GHG at a high spatial resolution of 10 km for Berlin to obtain anthropogenic $CO_2$ emissions, and to quantify the uncertainties in retrieved anthropogenic emissions related to instruments (e.g. CarbonSat) and modelling errors. In the present paper, our focus is on a high-resolution (1 km) study of both $CO_2$ and $CH_4$ in Berlin, and assess the performance of WRF-GHG through comparing the simulated wind and concentration fields to observations from wind stations and ground-based solar-viewing spectrometers. Then DCM is tested as a proper approach for model analysis, which can cancel out the bias from initialization conditions and highlight regional emission tracers. The simulation workflow is also adapted to this purpose where needed. This study is the fundamental study of the WRF-GHG mesoscale modeling framework in urban areas.

Line 125: One of the novelties of this work is to provide GHGs high-resolution simulations (1km x 1km). In this sense, it would be nice to discuss the impact of using, for example the EDGAR V4.1 inventory as anthropogenic fluxes at a spatial resolution of 0.1 (about 10 km x 10 km), on the simulated concentrations fields, or how the spatial resolution is treated in the VPRM model.

Response: Thank you for the valuable comment. The WRF model with finer resolutions is able to capture more details of local modifications to the simulated fields (e.g., wind fields (DuVivier, A. K., & Cassano, J. J., 2013)) and perform better than the coarse-resolution model for all simulations (Jee, J. B., & Kim, S., 2017). Compared to the coarse-resolution model, the fine-resolution WRF-GHG model in this study should be able to capture better meteorological fields and simulation fields which are suitable for comparing with the ground-based measurements.

In terms of the anthropogenic tracer, we use a quite simple method to downscale EDGAR inventory from 10 km to 1 km. Basically, each model grid is matched with the closest grid point in EDGAR inventory and saving the corresponding EDGAR emission flux to the initialization file of our model. During the simulation, this initialization file is read into the first layer of our domain. Then, combined with the meteorological conditions and boundary conditions for concentration fields, the data from this initialization file are used for the 'online' calculation, i.e. the WRF-GHG model.

For the biogenic tracer, in turn, VPRM calculates the hourly NEE based on MODIS satellite estimates of LSWI, EVI, etc. (Beck et al., 2011). Before using vegetation indices in WRF-GHG initialization, reflectance data from the MODIS satellite (8-day intervals and 500 m spatial resolution) is combined with Synmap vegetation classification (1 km resolution) and per vegetation class transformed to the desired projection (Lambert Conical Cartesian co-ordinate system in our case) within the WRF-VPRM preprocessor. A brief information on the treatment of anthropogenic and biogenic tracers has been included in Line 137-142:

The biogenic $CO_2$ emission is calculated online using VPRM (Mahadevan et al., 2008), in which the hourly Net Ecosystem Exchange (NEE) of $CO_2$ reflects the biospheric fluxes between the terrestrial biosphere and the atmosphere, estimated by the sum of Gross Ecosystem Exchange (GEE) and Respiration. VPRM in WRF-GHG calculates biogenic fluxes initialized by vegetation indices (land surface water index (LSWI), enhanced vegetation index (EVI), etc..) from the MODIS satellite (as available via https://modis.gsfc.nasa.gov/). Combined with SYNMAP vegetation classification at a resolution of 1 km, the refectance data from the MODIS satellite at a 500-m spatial resolution and 8-day intervals, is aggregated to the Lambert Conformal Conic (LCC) projection within the WRF-VPRM preprocessor. Then, the data including these high-solution vegetation indexes at a resolution of 1 km are available on the model domains.

Line 129: The authors mention that the time factors for the assimilated anthropogenic fluxes could introduce considerable uncertainties. Please, explain more in detail this issue, eg, the value of these uncertainties or how these could affect the GHGs concentration simulations.

Response: As documented in the EDGAR official website (https://themasites.pbl.nl/tridion/en/themasites/edgar/documentation/content/Temporal-variation.html), the temporal-variation set is originally made by Veldt (for the chemistry transport model (CTM) Long Term Ozone Simulation (LOTOS), a European climate model) and contains the variations for seasons (winter/summer), day-to-week (weekday/weekend) and day-and-night. This temporal variation set is defined based on Western European data. Thus, uncertainty can arise when applying this set to other European countries and even more so for other regions. Furthermore, the coarse anthropogenic-emission initialization in our study makes it hard to distinguish and locate the specific emission points within high-resolution model grids and weakens the impact from the real high emission hot-spot in urban area. We add information about these caveats in Line 149-153:

Here we apply time factors for seasonal, daily and diurnal variations defined by the time profiles published on the EDGAR website (http://themasites.pbl.nl/tridion/en/themasites/edgar/documentation/content/Temporal-variation.html); however, considerable uncertainties are to be expected in applying these time factors. This temporal variation set is derived based on western European data such that the representativity for other European countries and even other world regions may be quite poor. The coarse emission fluxes used for the initialization of the anthropogenic tracer in WRF-GHG can cause problems when locating emission points within the high-resolution model grid, and can weaken the impact from the real high emission hot-spots in the fine domain of our study.

Line 152. The field campaign is carried out between 23rd June and 11th July 2014. Why is the period from 1st July to 5th July only used for the wind comparison? Since EM27 observations are performed during day-time, it would be interesting to include the performance analysis for wind comparison distinguishing between day and nighttime.

Response: The simulation period for this Berlin case is from 1st July to 10th July. The figure which contains the comparisons of wind fields for ten days is too crowded. Figure 2 is updated already, and now includes wind comparisons for ten days. The reason why we do not include the period in June as our simulation period is that the measurement situations were not stable in June. Although the measurement campaign started from 23rd June, only the measured data for these 3 days (26th, 27th and 28th June) is provided (Hase et al., 2015) and the measurement qualities are not good (only 27th June is 'good', others are very poor). To be a fundamental study of the model framework which can run continuously and comparing the simulations with a relatively stable measurements, we decide to start our simulation run from the start of July.

We highlight the wind fields for the measurement periods (basically in the day time between 7 am to 16 pm), using the gray-shadowed squares in Fig.2 of the content. RMSE values are used to show the performance of wind fields. We conclude that the simulations fit better with the measurements in the day time, compared to the values in the nighttime. The measured wind fields in the nighttime holds larger variability compared with the simulation. All these contents are in Line 176-190:

Figure.2 shows the comparisons of wind speeds (Fig.2(a)) and wind directions (Fig.2(b)) between simulations and observations at 10 meters from 1st July to 10th July and the model-measurement differences. EM27/SUN only operates in the daytime under enough sunlight (the detailed description of the instrument can be found in Gisi et al.(2012), Frey et al.(2015) and Vogel et al.(2019)). The instrumental working periods are marked by gray shaded boxes in Fig.2. The measured (dashed lines) and simulated (solid) wind speeds (Fig.2(a)) at 10 meters show similar trends and demonstrate relatively good agreement over the 10-day time series with a root mean square error (RMSE) of $0.9247\,\mathrm{m/s}$. Large uncertainties in wind speeds are found to appear always with the lower wind speeds, mostly at night. In terms of wind directions at 10 meters, we observe that the simulated wind directions show similar but slightly underestimated fluctuations (Fig.2(b)), which result in a RMSE of $60.8328°$. Larger uncertainties in wind directions always exist during the low wind speed periods (Fig.2(a)&(b)). During the instrumental working period (within the daytime), the simulations fit better with the measurements with relatively lower RMSEs $0.6928\,\mathrm{m/s}$ for wind speeds and $41.4793°$ for wind directions. We find that the measured wind fields (both wind speeds and wind directions) have more fluctuations, compared to the simulations. This could be caused by real fast wind changes, which the model, simulating a somewhat idealized environment, is not able to capture. To be specific, local turbulence given by urban canopy, buildings etc. are not represented well in the model.

Line 159. Explain possible reasons for the discrepancies between the model and observed wind fields. Maybe the model

is not able to capture very fast changes of air masses, which will strongly affect the GHG comparisons.

Response: Thanks for your comment. The possible reason for the discrepancies between the simulated and measured wind fields have been added in the content (line 187-190).

Line 166. The influence of the EM27 vertical sensitivity is neglected in the model-observation comparison. But, the EM27 averaging kernels show a dependence on the solar geometry, which is considered in similar studies such as Vogel et al. (2018). Please justify more in detail why not to smooth the GHGs simulations (using the EM27 averaging kernels and a priori profile).

Response: Really thanks for your suggestion and we have added the smoothing part in the content (in Sect.3.3). We do agree with your opinion and smooth the simulation results, using the EM27 column sensitivities associated with solar zenith angels (SZA) and the a-priori $CO_2$ profile provided by the Whole Atmosphere Community Climate Model (WACCM) Version 6. Within such a limited simulation period in July (10 days) in our study, the daily variations of SZAs for these ten days are mostly overlapped (Fig.1(a) of this response). That is to say, we can use the simulation date with the most measurement data ($4^{th}$ July) as the reference for the column sensitivities with different SZAs. Then the column sensitivities following the model vertical pressure axes can be derived through interpolation on the basic of the reference column sensitivities.

As seen in Fig.1(b) and (c) of this response, the interpolated column sensitivities (the circles) fit well with the distribution of column sensitivities with different SZAs (the solid lines) for both $CH_4$ and $CO_2$. Compared to Vogel et al.(2018), we choose to interpolate the column sensitivities directly, instead of calculating the column sensitivities based on formulas, because the best measurement date in Berlin campaign ($4^{th}$ July) can provide us enough values to capture the vertical distribution features of column sensitivities accurately. The a-priori $CO_2$ and $CH_4$ profiles are provided by the Whole Atmosphere Community Climate model (WACCM) Version 6. The calculation details about the smoothed profiles for $XCO_2$ and $XCH_4$ have been rephrased in Sect.3.3 of the content and one example of this smoothing calculation is depicted in Fig.2 of this response for $XCH_4$ to better understand the smoothing process:

[Figure]

Figure 1: (a) The daily variations of solar zenith angles (SZA) for five simulation dates ($1^{st}$, $3^{rd}$, $4^{th}$, $6^{th}$ and $10^{th}$ July) and the vertical distributions of column sensitivities for (b) $CO_2$ and (c) $CH_4$ on $4^{th}$ July. (b) & (c): the solid lines represent the column sensitivities derived from EM27/SUN under different SZAs and the circles stand for the interpolated SZAs associated with the model pressure axes.

When comparing remote sensing observations to model data (or also datasets from different remote sensing instruments to one another), limitations of the instruments in reconstructing the actual atmospheric state need to be taken into account. In general, this requires the a-priori profile which was used for the retrieval and the averaging kernel matrix, which specifies the loss of vertical resolution (fine vertical details of the actual trace gas profile cannot be resolved) and limited sensitivity (e.g. Rodgers and Conner (2003)). In the case of EM27/SUN, the spectrometers used be the network offer

[Figure]

Figure 2: The calculations during the smoothing process. (a) the vertical concentration distribution derived from WRF-GHG output at 17 am on 4th July; (b) the solid line represents the column sensitivities derived from EM27/SUN and the orange dots stand for the interpolated column sensitivities associated with the model pressure axe; (c) the vertical concentration distributions weighted by the interpolated column sensitivities shown in (b); (d) the vertical distribution of concentrations derived from the priori model (WACCC); (e) the vertical distributions of the weighted (weight factors: 1-AK) priori concentrations; (f) the smoothed vertical concentrations profile obtained by the sum of (c) and (e).

only a low spectral resolution of $0.5\,\mathrm{cm}^{-1}$. Therefore, performing a simple least squares fit by scaling retrieval of the a-priori profile is generally appropriate. In this case, there is no need to specify a full averaging kernel matrix, instead, the specification of a total column sensitivity is sufficient. The total column sensitivity is a vector (being a function of altitude), which specifies to which degree an excess partial column superimposed on the actual profile at a certain input altitude is reflected in the retrieved total column amount. This sensitivity vector is a function of solar zenith angle (and ground pressure), mainly due to the fact that the observed signal levels in different channels building the spectral scene used for the retrieval are shaped by a mixture of weaker and stronger absorptions (if all spectral lines in the spectral scene would be optically thin and too narrow to be resolved by the spectral measurement, the sensitivity would approach unity throughout).

In order to ensure measurement qualities and enough sample points for further concentration comparisons, we select five measurement dates (1st, 3rd, 4th, 6th and 10th July) with relatively good measurement qualities (from fair '++' to very good '++++') based on Hase et al. (2015). The pressure-dependent column sensitivities for $CO_2$ (Fig.3(b)) and $CH_4$ (Fig.3(c)) are derived from measurements performed in Lindenberg on 4th July (the best measurement-quality day). Details about the measurements can be found in (Hase et al. (2015) and Frey et al. (2015). The shape and values of the column sensitivities from Karlsruhe closely resemble the results of Hedelius et al. (2016) in Pasadena. As depicted in Fig.3(a), the solar zenith angles (SZAs) are almost identical for each day in our study (at each hour), rendering the shape of column sensitivities (at a specific hour of the day) practically independent of the measurement date. The column sensitivities for 4th July (Fig.3(b,c)), are taken as a basis for our smoothing process below. The a-priori $CO_2$ and $CH_4$ profiles have been taken from the Whole Atmosphere Community Climate model (WACCM) Version 6 here. A smoothed profile for a target gas $G$ is then obtained as Eq.3 in (cf. Vogel et al., 2019),

$$G^s = K * G + (I - K) * G^p \tag{1}$$

where $G$ is the modelled profile from WRF-GHG, $I$ is the identity matrix, $K$ is a diagonal matrix containing the averaging kernel, and $G^p$ is the a-priori profile.

In order to compare the simulated smoothed concentration fields with the observations, the simulated smoothed pressureweighted column-averaged concentration for a target gas $G$ ($XG$) is calculated as,

$$\Delta p(i) = \frac{P(i) - P(i+1)}{P_{\text{sf}} - P_{\text{top}}} \rightarrow XG = \sum_{i=1}^{n} \Delta p(i) \times G^{s}(i) \tag{2}$$

Here, $\Delta p_i$ is the proportional to the differences of the pressure values $P(i)$ at the bottom and $P(i+1)$ at the top of the $i^{th}$ vertical grid cell; $P_{\text{top}}$ and $P_{\text{sf}}$ represent the hydrostatic pressures at the top and at the surface of the model domain, and $G^{s}(i)$ stands for the simulated concentration of the target gas $G$ at the $i^{th}$ vertical level.

In Figures.D1 and D2 of Appendix D, we compare the simulated $XCO_2$ and $XCH_4$ with and without smoothing. The simulated concentrations are only slightly enlarged after smoothing, approximately 1-2 ppm for $XCO_2$ and 2 ppb for $XCH_4$, while the variations are mostly not changed. Compare to the period with lower SZAs (at noon), the smoothed values in the morning and afternoon with higher SZAs hold relatively larger enlargements.

Line 176. For the comparison between simulated and observed column-averaged concentrations, the authors limit the analysis to the 3$^{rd}$ and 4$^{th}$ July. But, according to the wind comparison, the 3rd July shows the highest discrepancies between model and simulations. Therefore, observations and simulations could account for different air masses with different GHGs concentrations and influenced by different sources, which should be considered in the discussion. To account for this, the authors could plot the simulated-observed differences as a function of the wind differences. Although the EM27 stations do not coincide with the meteorological stations, figure 2 shows that the study area is very homogeneous with regards to wind fields. Thereby, these differences could be a proxy of the model inconsistences.

Adjust better the scales of figure 3 to clearer see the dispersion and distribution of the data, especially for 4$^{th}$. Now, it is hard to identify what database shows more variability, which could be interesting to know if the simulations are underestimating or overestimating the real CH4 variability.

Response: Thanks for your helpful suggestion! We have shown the concentrations for five targeted simulation dates and enlarged $CH_4$ concentrations to figure out the trends (details see Fig.3 and Sect.3.3). All the contents for the comparison of $CO_2$ and $CH_4$ have been somewhat re-worked. Now it can be seen that the measured $XCH_4$ holds larger variations. We also tried to figure out whether the bias is associated with wind differences, but there is no obvious connection between them.

Figure.4(a) shows the measured and modelled variations of $XCO_2$ and $XCH_4$ for these five days. Compared to the measurements, the smoothed simulated pressure-weighted column-averaged concentrations for $CO_2$ ($XCO_2$) show quite similar trends but with approx. 1-2 ppm bias, indicated by a RMSE of $1.2534$ ppm. The simulated $XCO_2$ are over-estimated for 1$^{st}$, 3$^{rd}$ and 4$^{th}$ July while on 6$^{th}$ and 10$^{th}$ July, the model is underestimated, which could attribute to the uncertainties from the coarse anthropogenic surface emission fluxes, background concentrations from CAMS (Sembhi et al., 2015), and the ignorance of the influence from the line of the sun sight.

Figure.4(b) shows the comparison of the pressure-weighted column-averaged concentrations for $CH_4$ ($XCH_4$) between observations and smoothed simulations on the five selected dates (1$^{st}$, 3$^{rd}$, 4$^{th}$, 6$^{th}$ and 10$^{th}$ July). We find that there is an approximate offset of 50-60 ppb between observations and models (RMSE = $58.1082$ ppb). The simulated $XCH_4$ is around 1860 ppb while the measured value is around 1810 ppb which is comparable to the values (1790-1810 ppb) observed at two Total Carbon Column Observing Network (TCCON) measurement sites in June and July 2014, Bremen in Germany (Notholt et al., 2014) and Bialystok in Poland (Deutscher et al., 2014). This bias of the simulated $XCH_4$ seems to be constant (around 2.7 %) each day. Thus, we introduce an offset applied to all sites for each simulation date to compare the model and the measured data, effectively removing the bias, which we attribute to a too high background $XCH_4$. The daily offset is assumed to be the difference between the simulated and measured daily mean $XCH_4$. After applying the daily offset, the measured $XCH_4$ shows a somewhat better agreement and a similar trend but with larger variability, compared to the simulation (RMSE = $3.1690$ ppb). The smaller variations from the simulation results can, e.g., be caused by the error from the spatial-temporal treatment of emission maps, underestimated emissions from anthropogenic activities, the coarse wind data and/or the smoothing of actual extreme values in the simulation.

Line 187-194. Although applying the offset seems to improve the comparison between observations and simulations, the model is not capturing well the observed $CH_4$ variability ($R^2$ is too small), thereby model and observations are not reflecting the same air masses, sources (industries or natural processes). The authors superficially mention the possible influence of the tropopause height in the simulations, but without quantifying this impact. Have the authors considered the possible influence of the PBL? Or the shape of the constant a priori profile used for EM27 retrievals? Please include

a more detailed discussion of the possible reasons for these discrepancies.

Response: Thanks for your valuable comments. For this $CH_4$ bias, we attribute to the errors in the troposphere height and the general offset from CAMS dataset in Sect.3.3. For the vertical distribution of our case, the upper pressure is up to 50 hPa and the simulated RBL height varies from approx. 200 m (in the morning and at night) to 1500 m (during the daytime). And we use the vertical distribution of water vapour to check the simulated PBL height and conclude that the PBLs are situated in the right vertical layers. With the restriction of the relatively coarse vertical layers (26 layers) in our vertical distribution, the PBL heights potentially hold some uncertainty. The $CH_4$ vertical profile changes with the increase of PBL heights, which leads to the error in $XCH_4$. The constant a priori profile shapes in EM27/SUN retrievals (while the actual atmospheric profiles are variable) do attribute to the smoothing error of measurement values.

A major offset in modelled $CH_4$ concentration fields could potentially be attributed to the errors in the troposphere height and a general offset from CAMS. In the $CH_4$ vertical concentrations profile, we find that the typical sharp decrease is given within the tropopause height. Tukiainen et al. (2016) also find the similar sharp decrease when using the air-cores to retrieve the atmospheric $CH_4$ profiles in Finland. During the simulation, the background concentration values of CAMS are directly fitted to the WRF pressure axis, without the consideration of the actual tropopause height, thus this is unlikely to be the case. An illustration of the vertical distribution for $CH_4$ is provided in Appendix C. While the $CO_2$ vertical distribution shows a quite flat decrease with the increase of pressure and there is no need to consider the tropopause height during the grid treatment in the vertical layer. Then in view of CAMS, the reports from Monitoring Atmospheric Composition and Climate (MACC) described that CAMS held an increase of bias and RMSE (approx. 50 ppb) in each part of the world, compared to the Integrated Carbon Observation System (ICOS) observations in 2017 (Basart et al., 2017). Galkowski et al (2019) also mentioned one $CH_4$ offset (approx. 30 ppb within troposphere), when initializing the concentration fields using CAMS. Apart from these two major potential reasons for the bias, the influence from the in-accurate simulated planetary boundary layers and the shape of the constant a priori profile used for the retrievals both potentially contribute to the discrepancies for the concentration fields. Due to the lack of fine measured vertical concentration profiles, it is not easy to quantify these errors and attribute these potential reasons to this 2.7% error quantitatively. Thus, a DCM-based analysis is presented in Sect.4, aiming at eliminating the bias from these relatively high initialization values for $CH_4$ and making it easier to assess WRF-GHG results with respect to the measurements.

Line 195-235: As mentioned in the general comments, the discussion and attribution of GHGs sources is very simple. Please consider to include a robust statistics analysis to support the main conclusions of this section.

Thanks for your suggestions. In this study, our focus is on the model for the major emission sources. We would like to know the major contributions for urban areas and test whether the DCM can work well to highlight the influence from human activities and weaken the impact from the strong biogenic emissions. Thus, the main conclusions for the tracer analysis are demonstrated (Line 370-380) that 'The biogenic component plays a pivotal and leading role in the variations of $XCO_2$. The impact from anthropogenic emission sources is somewhat weak compared to this, while for $XCH_4$, the enhancement is dominated by human activities.' The WRF-GHG mesoscale modelling framework is being built in Munich based on the first worldwide permanent column measurement network of Munich. The quantification analysis for different emission tracers is suggested to run for this Munich in which more emission tracers (e.g., biogenic emissions from wetland for $XCH_4$, the traffic emission and strong point sources' emissions in urban areas, parts of the anthropogenic emissions) are being added and more data are available. In this Munich case, it is possible to get the long-period data and we can do the 'real' simulation for anthropoogenic emission, instead of the simulated values entirely based on the priori data.

Line 248. Have the authors analyzed the vertical distribution of the winds within PBL for the comparisons?

The vertical layers in WRF model are defined following the pressure definition and the top-layer pressure (50 hPa) is already up to tropopause. PBL normally varies from a few hundred meters (morning and night) to over one thousand meters (daytime, pick at noon) which are situated from the second vertical layer (morning and night) to $13^{th}$ layer (noon) of our domain, corresponding (see the left column of Fig.3 below). The wind is higher than the surface wind. The WRF-GHG outputs do provide these simulated PBL values but there is lack of observations to assess these simulated values. Avolio et al., (2017) did some sensitivity analysis of PBL in the WRF model with a case study of southern Italy, and concluded that the simulated PBL heights are mostly overestimated.

Line 260-289. A plot showing the CH4 and CO2 enhancement observed and simulated as a function of the wind

[Figure]

Figure 3: Variations of the simulated Planetary Boundary Layer (PBL) Height (left side), and the wind speeds (middle) and wind directions (right) within PBL on July $3^{rd}$ July in Tegel. The colors from black to blue represent the time from morning to evening. The bold solid lines represent the values within PBL.

directions or differences between simulated and observed wind directions could help to explain better the results of the section 4. Regarding to Figure 7 and 8, why are not the wind fields considered in Figure 7 similarly to Figure 8? Why does not Figure 8 include the 1st July? Why are not the performance values for $CO_2$ included in the text?

Thanks for your suggestions. We did several tests to find whether the biases of $\Delta XCO_2$ and $\Delta XCH_4$ between measurements and simulations are associated with wind fields or differences between measured and simulated wind directions. A dependency on wind speeds or wind directions is not found in this study (like the scatter-plot (Fig.9) provided by Vogel et al., (2019)). But in our case, we find that the extreme comparison points in the right column of Fig.8 always correspond to higher discrepancies of wind directions between measurements and simulations. These contents are included in Sect.4.1 and the Fig4 and Fig.5 of this response (Fig.7 and Fig.8 of the content):

To further understand the differences of $\Delta XCO_2$ and $\Delta XCH_4$ between measurements and simulations (see Fig.7, right column and Fig.8, left), the comparison of hourly mean $\Delta XCO_2$ and $\Delta XCH_4$ values for these four targeted dates is illustrated in the right column of Fig.8. Due to the restriction of measured wind information, we illustrate the difference of simulated and measured wind directions at 10 meters (i.e. Fig.2(b)) with respect to the hourly mean $\Delta XCO_2$ and $\Delta XCH_4$. We find that the real hourly mean $\Delta XCO_2$ and $\Delta XCH_4$ values are generally higher than the simulated values. Extreme points are colored by red and blue in the right column of Fig.8, standing for large differences between measured and simulated wind directions at 10 meters. We see that a large difference of wind directions is a necessary but insufficient condition for the bias of $\Delta XCO_2$ and $\Delta XCH_4$ between measurements and simulations. In future studies, this may be verified further.

Then to make the consistence with targeted simulation dates for each comparison case, we target $3^{rd}$, $4^{th}$, $6^{th}$ and $10^{th}$ July in this case (see Fig.7, Fig.8, Fig.9 and Fig.10, Sect.3.3 and Sect4).

[Figure]

Figure 4: Modeled wind fields for downwind (blue lines) and upwind (red) sites (left column), and downwind-minus-upwind differential evaluation for measured (blue) and simulated (black) $XCH_4$ (right column) on $3^{rd}$, $4^{th}$, $6^{th}$ and $10^{th}$ July 2014. Based on the selection of downwind and upwind sites in Table.1, $\Delta XCH_4$ is calculated using Eq.6, 7 and 8, depicted by blue lines for measurements and black lines for simulations. The black error bars in the right column are the standard derivations of the minute values of the hourly mean.

[Figure]

Figure 5: Measured (black lines) and simulated (blue) $\Delta XCO_2$ on $3^{rd}$, $4^{th}$, $6^{th}$ and $10^{th}$ July 2014, and Comparison of hourly mean $\Delta XCO_2$ and $\Delta XCH_4$ for these four days. The $\Delta XCO_2$, calculated using Eq.5, 6 and 7, are depicted by blue lines in the right column. The red and green lines show the variation of the differences between downwind and upwind sites in $XCO_2$ changes from anthropogenic and biogenic activities, respectively. The points color in right column are coded by the difference of the simulated and measured wind directions at 10 meters.

**References**

Avolio, E., Federico, S., Miglietta, M. M., Feudo, T. L., Calidonna, C. R., Sempreviva, A. M.: Sensitivity analysis of WRF model PBL schemes in simulating boundary-layer variables in southern Italy: an experimental campaign. Atmospheric research, 192, 58-71, https://doi.org/10.1016/j.atmosres.2017.04.003, 2017.

Beck, V., Koch, T., Kretschmer, R., Marshall, J.and Ahmadov, R., Gerbig, C., Pillai, D., and Heimann, M.: The WRF Greenhouse Gas Model (WRF-GHG). Technical Report No. 25, Max Planck Institute for Biogeochemistry, Jena, Germany., 2011.

Deutscher, N. M., Notholt, J., Messerschmidt, J., Weinzierl, C., Warneke, T., Petri, C., Grupe, P., and Katrynski, K.: TC-CON data from Bialystok (PL), Release GGG2014R1, TCCON data archive, hosted by CaltechDATA, https://doi.org/10.14291/tccon.ggg2014.bialystok01.R1/1183984, https://tccondata.org, 2014.

DuVivier, A. K., Cassano, J. J.: Evaluation of WRF model resolution on simulated mesoscale winds and surface fluxes near Greenland. Monthly Weather Review, 141(3), 941-963, https://doi.org/10.1175/MWR-D-12-00091.1, 2013.

Gierczak, T., Talukdar, R. K., Herndon, S. C., Vaghjiani, G. L., Ravishankara, A. R.: Rate coefficients for the reactions of hydroxyl radicals with methane and deuterated methanes, The Journal of Physical Chemistry A, 101(17), 3125-3134, https://doi.org/10.1021/jp963892r, 1997.

Hardiman, B. S., Wang, J. A., Hutyra, L. R., Gately, C. K., Getson, J. M., and Friedl, M. A.: Accounting for urban biogenic fluxes in regional carbon budgets, Science of the Total Environment, 592, 366–372, https://doi.org/10.1016/j.scitotenv.2017.03.028, 2017.

Hase, F., Frey, M., Blumenstock, T., Groß, J., Kiel, M., Kohlhepp, R., Mengistu Tsidu, G., Schäfer, K., Sha, M., and Orphal, J.: Application of portable FTIR spectrometers for detecting greenhouse gas emissions of the major city Berlin, Atmospheric Measurement Techniques, 8, 3059–3068, https://doi.org/10.5194/amt-8-3059-2015, 2015.

Jee, J. B., Kim, S.: Sensitivity study on high-resolution WRF precipitation forecast for a heavy rainfall event. Atmosphere, 8(6), 96, https://doi.org/10.3390/atmos8060096, 2017.

Notholt, J., Petri, C., Warneke, T., Deutscher, N. M., Buschmann, M., Weinzierl, C., Macatangay, R., and Grupe, P.: TC-CON data from Bremen (DE), Release GGG2014R0, TCCON data archive, hosted by CaltechDATA, https://doi.org/10.14291/tccon.ggg2014.bremen01.R0/1149275, https://tccondata.org, 2014.

Pillai, D., Buchwitz, M., Gerbig, C., Koch, T., Reuter, M., Bovensmann, H., Marshall, J., and Burrows, J. P.: Tracking city $CO_2$ emissions from space using a high-resolution inverse modeling approach: a case study for Berlin, Germany, Atmospheric Chemistry and Physics, 16, 9591–9610, https://doi.org/10.5194/acp-16-9591-2016, 2016.

Ridgwell, A. J., Stewart J. M. Keith G.,: Consumption of atmospheric methane by soils: A process-based model. Global Biogeochemical Cycles, 13.1, 59-70, https://doi.org/10.1029/1998GB900004, 1998.

Vogel, F. R., Frey, M., Staufer, J., Hase, F., Broquet, G., Xueref-Remy, I., Chevallier, F., Ciais, P., Sha, M. K., Chelin, P., Jeseck, P., Janssen, C., Té, Y., Groß, J., Blumenstock, T., Tu, Q., and Orphal, J.: $XCO_2$ in an emission hot-spot region: the COCCON Paris campaign 2015, Atmos. Chem. Phys., 19, 3271-3285, https://doi.org/10.5194/acp-19-3271-2019, 2019.

---

## Author Comment (AC4) · 15 May 2019

We thank Reviewer #2 for helpful and valuable comments. The rephrased manuscript is uploaded as the supplement.

Please also note the supplement to this comment:
https://www.atmos-chem-phys-discuss.net/acp-2018-1116/acp-2018-1116-AC4-supplement.pdf